# Heterogeneous Transfer Learning for Wi-Fi Indoor Positioning Based Hybrid Feature Selection

**DOI:** 10.3390/s22155840

**Published:** 2022-08-04

**Authors:** Hailu Tesfay Gidey, Xiansheng Guo, Lin Li, Yukun Zhang

**Affiliations:** 1Department of Information and Communication Engineering, University of Electronic Science and Technology of China, Chengdu 611731, China; 2Yangtze Delta Region Institute (Quzhou), University of Electronic Science and Technology of China, Quzhou 324000, China

**Keywords:** indoor positioning, hybrid feature selection, heterogenous transfer learning, CRLB analysis, optimization

## Abstract

This paper presents the application of heterogeneous transfer learning (HetTL) methods which consider hybrid feature selection to reduce the training calibration effort and the noise generated by fingerprint duplicates obtained from multiple Wi-Fi access points. The Cramer–Rao Lower Bound analysis (CRLB) was also applied to evaluate and estimate a lower limit for the variance of a parameter estimator used to analyze positioning performance. We developed two novel algorithms for feature selection in fingerprint-based indoor positioning problems (IPP) to enhance positioning performance in the target domain with the HetTL. The algorithms comprised two scenarios: (i) a principal component analysis-based approach (PCA-based) and (ii) a hybrid approach that takes both PCA and correlation effect analysis into account (hybrid scenario). Accordingly, a new feature vector was constructed by retaining only the most significant predictors, and the most efficient feature dimensions were also determined by using a hybrid-based approach. Experimental results showed that the hybrid-based proposed algorithm has the minimum mean absolute error. The CRLB analysis also showed that the number of Wi-Fi access points could affect the lower bound location estimation error; however, identifying the most significant predictors is an effective approach to improve positioning performance.

## 1. Introduction

Indoor positioning systems (IPSs) have received significant attention in both academia and industry due to the growth of wireless communication infrastructure and the ever-increasing demand for location-based services (LBSs) [1,2,3,4]. Currently, more than 80% of the world’s population owns a smartphone [5], it is estimated that most people spend around 80% of their daily lives indoors [6], and 74% of smart device owners are active users of smartphone location-based applications [7,8]. The predominant use of location-based information is required for a variety of applications, including but not limited to: military use (originally designed for this purpose based on Global Positioning System (GPS)), shopping malls to guide customers to obtain services, hospitals to monitor patients for better health services, marketing to assist in the display of advertisements, emergency services, navigation, social networking services, multimedia geotagging, location and tourism, and so on [1,7,8,9,10,11,12,13]. Although GPS receivers are known for their promising and dependable accuracy in an outdoor environment, their applications are limited to indoor positioning due to the complex nature of the indoor environment, which includes: no direct line of sight, poor GPS signal penetration through complex internal buildings, and severe internal channel conditions such as shadows and multipath fade [14,15]. In addition to higher accuracy, an indoor positioning system for mobile devices should have low computational complexity and a short positioning time, necessitating the use of alternative wireless indoor positioning technologies.

Aside from that, most services are location-based, and with the rapid rise in people’s daily lives that happen to be indoors, a new approach to location systems that utilize existing wireless communication infrastructure that is economically viable is required. To address indoor positioning problems, several measurement methods have been proposed and studied in the literature, including time of arrival (TOA) [16], angle of arrival (AOA) [17], channel state information (CSI) [18], and power of the received signal (RSS) [19,20,21,22,23,24,25]. Because they require additional hardware devices to store accurate timing information and match signal directions, the first three methods cannot be used directly with the existing wireless network or require high construction costs [25,26,27]. The first method (TOA/TDOA) specifically requires precise timing information, which is difficult to obtain in a complex multipath indoor environment, and non-line-of-sight (NLOS) is also a challenge [25]. Second, the AOA method requires additional hardware devices that could make it suitable for signal directions [26]. Similarly, indoor positioning based on CSI needs extra hardware known as Wi-Fi network interface cards (NICs) to extract CSI measurements from Wi-Fi networks [27] that can be used as fingerprints for positioning users or mobile devices. However, the RSSI-based method can be implemented directly using the existing wireless communication infrastructure without additional hardware devices. Received Wi-Fi signal strength (RSSI) (measured in decibels milliwatts: dBm) is used to find a relationship between transceivers or to measure location accuracy based on distances between the mobile user and Wi-Fi access points (Wi-Fi APs)available [3,28,29,30].

Thus, in this paper, the Wi-Fi RSS fingerprinting-based indoor positioning was used because it is the most widely used technique due to its low cost and ease of implementation [28,29,30], which consists of two stages: training and testing phases. During the training phase, RSS fingerprints (also known as radio maps) are collected from each Wi-Fi access point at multiple locations within the defined Grid Points (GPs), and a predictive model is trained to determine the “signal-to-location” learning relationship (offline phase). The learned model is then used during the test phase to deduce the location of the mobile user based on the new measurement received (online phase) [28,29,30]. Despite this approach’s popularity and low cost, the scattering of RSS values could be easily affected by multipath diffraction, as shadowing and scattering are common indoors [31,32,33]. Moreover, signal strength attenuation in wireless communications can occur due to various predictive factors such as path loss, shading, and multipath fading [34]. Furthermore, to achieve the desired accuracy with RSS-based fingerprinting, a sufficient number of labeled samples must be collected, which is both expensive and time-consuming.

Several Wi-Fi-RSS-based methods have been proposed and investigated to manage these complex indoor fluctuations and other challenges. For instance, the path loss model has been used to model signal jitter, but its application is limited due to the need for map information and a fixed receiver direction [35,36]. Various machine learning (ML) algorithms were also used to address the RSS fingerprint-based indoor positioning problems, but they did not take into account potential implications that could influence location estimation, such as incorporating related source domain into the target domain to improve positioning performance [37,38,39,40]. Although the scope of our work is limited to a positioning system-based single signal feature, hybrid methods such as the hybrid location system of various location applications (an integral of Bluetooth, Wi-Fi, UWB, and ZigBee) [41], Wi-Fi with Visual Light Positioning (VLP) [42], Wi-Fi beacons and Bluetooth Low Energy (BLE) [43], and others have been proposed to improve indoor positioning performance. Transfer learning (TL) methods, however, are effective for many real-world applications in the literature, as they leverage the knowledge present in labeled training data from a source domain to improve the model’s performance in a target domain with little or no labeled target training data [44]. The primary goals of this paper are to improve the positioning performance of Wi-Fi RSS fingerprinting-based indoor positioning by reducing computational complexity both in terms of cost and time through the use of heterogeneous transfer learning (HetTL) based hybrid feature selection. Moreover, this paper attempts to estimate the lower bound variance for the estimator through Cramer–Rao lower bound analysis.

However, there are two major challenges associated with HetTL techniques: (a) Feature spaces’ dimensions can vary and are inherently heterogeneous. (b) Multiple Wi-Fi access points may duplicate RSS fingerprints, resulting in interference patterns that match the user’s true fingerprint, or irrelevant features may be generated and saved in the database, inflating or degrading positioning performance. In other words, multiple Wi-Fi access points can emit signals over time, causing a single fingerprint to represent multiple reference points (fingerprint duplication), or the distribution of signals at a reference point can vary significantly, resulting in mismatch fingerprinting that is inconsistent with the user’s actual fingerprint (fingerprint mismatching). In this paper, the contributions are:(1)We proposed the combination of both PCA and Heterogeneous Transfer Learning methods to minimize fingerprint duplications by constructing a new feature vector that could extract the most significant features as well as enable positive knowledge transfer for the location estimation.(2)We applied correlation analysis (CA) to test the independence of the Wi-Fi received signals from multiple access points mainly to reduce the effect of multicollinearity among the various RSS Wi-Fi access points aiming to avoid irrelevant features from the analysis. We claim that the RSS fingerprints collected from a grid point might be duplicated possibly from multiple Wi-Fi APs and create interference of matched patterns with the actual user’s fingerprint or irrelevant features might be generated and recorded in the database which negatively inflates or degrades the positioning performance.(3)We build a fusing coefficient approach to exploit both offline and online sources to reduce labor-intensive database fingerprint construction and enhance positioning performance in the target domain through heterogenous transfer learning-based hybrid feature selection.(4)We analyze the unbiasedness of the location estimator for the fingerprint-based indoor positioning system or estimate lower bound variance for the estimator through Cramer–Rao lower bound analysis.

The remainder of this paper is structured as follows: related works are presented in Section 2. Section 3 discusses the fingerprinting positioning framework, as well as its problem formulation, evaluation matrices, and Cramer–Rao lower bound analysis (CRLB). Section 4 presents experimental results and discussions. Conclusions are provided in Section 5.

## 2. Related Works

This section provides an overview of indoor positioning systems, as well as potential location estimation performance evaluation metrics and the challenges that limit the applications of Wi-Fi RSS signal features in indoor positioning. Indoor positioning systems have attracted a significant amount of research effort in the last two decades due to the increasing use of location-based services (LBSs) that happen to be mostly indoors. Due to the poor performance of GPS signals indoors, various wireless technologies have been proposed and studied to solve indoor positioning problems with greater accuracy [45,46,47]. Despite advances in the location service industry over the past decades, indoor positioning remains a more complex task than outdoor positioning due to the complex nature of the indoor environment, where radio signals are characterized by NLOS propagation, multipath effects, and a dynamic environment [45,46,47,48]. Thus, an indoor positioning system should not only have higher positioning accuracy, but also low computational complexity and a short positioning time. To this end, a novel approach to indoor positioning systems that takes advantage of existing wireless communication infrastructure and is economically viable is also required.

Several signal features, including channel state information (CSI), angle of arrival (AOA), time of arrival (TOA), and received signal strength (RSS), have been used as fingerprints to estimate the location of a user or mobile device to address indoor positioning requirements [15,20]. However, the first three measurement methods (TOA, CSI, and AOA) cannot be used directly with the existing wireless network infrastructures and have high implementation costs because they require additional hardware devices to store accurate timing information, network interface cards (NICs) to extract CSI measurements, and multi-directional antennas [25,26,27]. In contrast, the RSS-measurement method has been widely used as a location fingerprint for indoor positioning due to its low cost and ease of implementation [3,28,29,30]. There are two types of RSS-based localization methods: (a) propagation-model-based and (b) fingerprint-based [49]. The propagation model method is less robust in general because it requires prior knowledge of the locations of the Wi-Fi access points, and the RSS distributions can deviate dramatically in complex indoor environments [50]. The fingerprint-based approach [28,29,30,51,52,53] is, however, relatively simple, robust, and widely used for indoor localization, and is divided into two phases: (a) training and (b) testing. During the training phase, a survey is conducted to collect RSS measurements from predetermined known locations, and a database of fingerprint patterns indexed with their corresponding grid points (GPs) is created. During the testing phase, the learned model is used to deduce the location of the mobile user based on the new measurements received (online phase). Despite RSS’s popularity and low cost, multipath diffraction, shadowing, and scattering, which are common in indoor environments, can easily affect the spread of RSS values. Furthermore, to achieve the desired accuracy, RSS fingerprinting requires a large number of labeled samples, which is both labor-intensive and time-consuming.

Several research works have been proposed to deal with the complex nature of indoor environments causing poor indoor positioning performance [54,55,56,57,58,59,60,61]. For instance, some studies have been reported in the literature that use a probabilistic approach to address the dynamic signal fluctuations that are generated within complex indoor environments [56,57,58,59,60,61]. The probability models, however, are only valid or provide reliable results when certain conditions are met by the measurements; otherwise, they are no longer valid or superior. The assumption of normal distribution of signal values in a wireless indoor environment seemed impractical for various reasons including measurement time differences, the requirement for large instances, hardware devices (even identical mobile devices could follow different patterns), user orientations, inherent heterogeneity, and others [56,57,58,59,60,61]. Various machine learning algorithms were also widely used in the literature to address indoor positioning systems [37,38,39,40,41]. These systems seek the best possible match between the user’s fingerprint and a pre-defined set of grid points on the radio map. Fingerprints, however, may be duplicated from the available Wi-Fi Access Points (Wi-Fi APs) and interference, increasing the number of matched patterns with the user’s fingerprint (causing irrelevant features). In summary, the following are the two main types of localization algorithms used for RSS fingerprint-based indoor positioning systems: probabilistic and deterministic techniques, and their details are given below.

### 2.1. Deterministic Method

In deterministic technique, the average value of the Wi-Fi received strength signals collected on training phase from each access point (AP) at a particular grid point (GP) is used to create the database fingerprint and mainly we used this single value representing the distribution of an access point to map with the actual RSS measurement to complete the location estimation. In this method, the location of a mobile user ($\hat{L}$) is estimated as a convex combination of the *K* reference locations [62,63,64]. $\hat{L}$ is calculated as the shortest distance between $S_{ij}^{\left(g\right)}$ and ${\hat{S}}_{ij}^{\left(g\right)}$ in the *n*-dimensional space by using the following equation:(1)$$\hat{L}=\sum_{i=1}^{K}\left(\frac{\beta_{i}}{\sum_{j=1}^{K}\beta_{j}}\left\|S_{ij}^{\left(g\right)}-{\hat{S}}_{ij}^{\left(g\right)}\right\|\right)$$

The norm vector space given in Equation (1) can be rewritten as ${L^{\prime }}_{i}=\left\|S_{ij}^{\left(g\right)}-{\hat{S}}_{ij}^{\left(g\right)}\right\|$ such that $\left\{{L^{\prime }}_{1},\;{L^{\prime }}_{2},\;\ldots ,\;{L^{\prime }}_{k}\right\}$ represents the target locations between the corresponding matching fingerprint ${\hat{S}}_{ij}^{\left(g\right)}$ and the new received measurement at testing phase $S_{ij}^{\left(g\right)}$. This deterministic method is widely used in Wi-Fi indoor positioning systems due to its ease of use in both the creation of the fingerprint database and the localization, which is based on the average value of the received signal strengths of Wi-Fi access points at a specific GP. For example, a radio-frequency (RF)-based system based on the K-nearest neighbor (KNN) algorithm was proposed and studied for positioning and tracking users inside buildings, and the experimental results revealed that Radar can estimate user location with a high degree of accuracy [65]. Furthermore, RSS fingerprint-based ML algorithms such as k-nearest neighbor [65,66,67,68], radial basis function [69], weighted k-nearest neighbor [65,66], support vector machine [70], neural networks [71], and others such as kernel direct discriminant analysis and relevance vector regression [72] are being used for indoor positioning systems. Nonetheless, the spread of RSS values measured at a specific grid point is so dynamic in nature due to the inherent heterogeneity of hardware devices and time differences in measurements that there is no guarantee that the signal fluctuations will be represented by a single value for a specific position even with the same device. This would lead us to another approach known as the probabilistic approach, which could achieve better accuracy by taking signal distributions into account [73,74].

### 2.2. Probabilistic Techniques

In the probabilistic method, location $L_{i}$ is estimated by calculating and maximizing the conditional posterior probabilities $p\left(L_{i}|s\right),\;i=1,\;\ldots ,\;K$ given an observed fingerprint *s* and a fingerprint database. A Bayesian approach for Wi-Fi localization [56,57,58,59,60,61,63] can be given as:(2)$$p\left(L_{i}|s\right)=\frac{p\left(s|L_{i}\right)p\left(L_{i}\right)}{p\left(s\right)}$$
where $p\left(L_{i}|s\right)$ is the probability of a mobile user being at a location given that the new received signal strength, $p\left(s|L_{i}\right)$ is the probability of a Wi-Fi received signal strength of being located at location $L_{i}$ and can be calculated using the fingerprint database. The occurrence of each RSS value is used to generate a probability distribution as the likelihood function. Additionally, $p\left(L_{i}\right)$ is the a priori probability calculated or set based on researchers’ knowledge of the target’s characteristics. p(s) is a normalizing constant that does not depend on the location and is given as $\sum_{i=1}^{K}p\left(s|L_{i}\right)p\left(L_{i}\right)$. Probabilistic models are valid or would produce reliable results under the specific conditions that the measurements must meet, or their results are no longer valid or better. Furthermore, the signal spreads of the RSS values used for indoor positioning systems appear to be difficult to represent with a single value due to the complex nature of the indoor environment, where distributions of signals at a specific location even with the same device appeared to obey a specific pattern, making it difficult for the probabilistic approach to be efficient for indoor positioning [61]. Furthermore, in [61], the Gaussian Mixture Model (GMM) was reported as the highest mis-locating classifier for a wireless mobile user in an indoor environment. Several attempts have been made in the literature to solve the underlying assumption associated with probabilistic models, including transfer learning (TL) and taking significant instances during the training phase to establish a fairly stable ‘signal-signature to a location’ relationship, despite its criticism for being a costly and labor-intensive approach. Nonetheless, due to the requirements for a large sample size, these rewards based on a probabilistic approach could be obtained at the expense of computational cost and memory usage.

## 3. Problem Formulation and Framework

The Wi-Fi received strength signal fingerprint-based positioning is identified as the most popular indoor positioning technique due to its low cost and ease of implementation, which consists of two phases: (a) training and (b) testing. During the training phase, RSS fingerprints known as radio maps are collected from each WiFi access point at multiple locations within the defined space of grid points (GPs), and a predictive model is trained on the RSS measurements of a grid point possibly from all Wi-Fi access points to characterize the ‘signal–location’ relationship, and its corresponding index of the location is stored in the database fingerprints (offline phase). This process is repeated for all GPs in order to characterize or store a grid point’s signal signature with its corresponding location. The learned model is then applied during the testing phase to infer the mobile user’s location based on the new measurement by mapping into the highest likelihood resemblance of signal-signature stored during the training phase time (online phase). To formulate the problem mathematically, let us conceive the general fingerprint-print-based positioning of the indoor environment scenario partitioned into G grid points. Each represent a mobile user’s location and each GP is indexed with a label g, (g=0, …, k−1), M detectable Wi-Fi access points (j) were available, and the *i*th Wi-Fi signal strength of fingerprints received at the *k*th gird point of the *j*th access point is a vector and the fingerprint database can be represented as a matrix $F_{ij}^{\left(g\right)}$:
(3)$$F_{ij}^{\left(g\right)}=\left[\begin{array}{c} \begin{array}{ccc} ss_{11}^{\left(0\right)} & \cdots & ss_{1m}^{\left(0\right)}\\ \vdots & \ddots & \vdots \\ ss_{n1}^{\left(0\right)} & \cdots & ss_{nm}^{\left(0\right)} \end{array}\\ \begin{array}{ccc} ss_{11}^{\left(1\right)} & \cdots & ss_{1m}^{\left(1\right)}\\ \vdots & \ddots & \vdots \\ ss_{n1}^{\left(1\right)} & \cdots & ss_{nm}^{\left(1\right)} \end{array}\\ \begin{array}{ccc} ss_{11}^{\left(k-1\right)} & \cdots & ss_{1m}^{\left(k-1\right)}\\ \vdots & \ddots & \vdots \\ ss_{n1}^{\left(k-1\right)} & \cdots & ss_{nm}^{\left(k-1\right)} \end{array} \end{array}\right]$$

The above matrix can be described explicitly as: $F_{ij}^{\left(g\right)}=\left\{F_{ij}^{\left(g\right)};\;\left(x_{g},y_{g}\right)\right\};\;i=1,\;2,\;\ldots ,\;n\;\&\;j=1,\;2,\;\ldots ,\;m$ and $\left(x_{g},y_{g}\right)$ is the corresponding coordinate to the associated location of the signal signature. The target instances would be received during the testing phase denoted as $\left\{F_{ij}^{\left(g\right)}\right\};\;i=1,\;2,\;\ldots ,\;n_{t}\;\&\;j=1,\;2,\;\ldots ,\;m_{t}$ and the offline source domain can be represented as: $F_{ij}^{\left(g\right)}=\left\{F_{ij}^{\left(g\right)};\;\left(x_{g},y_{g}\right)\right\};\;i=1,\;2,\;\ldots ,\;n_{s}\;\&\;j=1,\;2,\;\ldots ,\;m_{s}$ called the labeled source data. $n_{t}$ and $n_{s}$ represent the numbers of measurements for the target and source data instances, respectively. In summary, the major challenges for Wi-Fi RSS-based fingerprint applications for indoor positions can be divided into three categories: (a) The inherent heterogeneity in measurement distribution at specific grid points or the dynamic nature of RSS value distributions (the sources of spreads can be: hardware devices, measurement time differences, user and antenna orientation, multipath channel conditions such as diffraction, shadowing, fading, interference, and others). (b) RSS fingerprints for a grid point may be duplicated from multiple Wi-Fi APs, causing interference of matched patterns with the actual user’s fingerprint, or irrelevant features may be generated and recorded in the database, inflating or degrading positioning performance. (c) Spending a significant amount of time during the training phase to establish a fairly stable ‘signal-signature to a location’ relationship, or their requirement for large sample size is so expensive. In this paper, we proposed a novel approach to solving the underlying problem associated with signal fluctuations by combining both techniques of principal component analysis (PCA) and correlation analysis with the help of heterogeneous transfer learning (HetTL), such that positive knowledge from the training phase could be leveraged, and only the most significant features would be considered to build the model, and we could establish fairly stable ‘signal-signature to a location’ relationship at the testing phase, respectively. Figure 1 demonstrates the proposed framework of Wi-Fi received signal strength-based fingerprint for the indoor positioning system.

### 3.1. Principal Component Analysis (PCA)

Even though the Wi-Fi received signal strength fingerprint-based technique has been identified as the most popular technique for indoor positioning systems due to the use of existing wireless infrastructure, its applications continue to be challenged. This is primarily because the spread of RSS values measured at a specific grid point is so dynamic, and the inherent heterogeneity of hardware devices, as well as time differences in when measurements were taken, have a significant effect on the distribution of RSS values received at each grid point, such that there is no guarantee that the signal fluctuations will be represented by a single value for a specific position, even when using the same device. To address these issues, we proposed combining PCA and HetTL techniques to create new feature spaces that may extract the most significant features while also enabling positive knowledge transfer for location estimation. Second, the algorithm must be computationally efficient while remaining reasonably priced. As a result, the PCA was used to reduce high-dimensional feature spaces to low-dimensional feature spaces by taking only the most significant features, although discarding information could negatively affect the ability to represent the entire measurements [75,76]. The PCA is an unsupervised learning technique for reducing data dimensionality to find the significant features in a dataset while retaining a large amount of information and increasing data interpretability [75].

The proposed algorithm (feature selection based PCA or Algorithm 1) is a pseudo-code used to generate a new fingerprint feature space based on PCA. The input values are the original database fingerprints; after PCA is applied, a new fingerprint feature space with reduced dimension is created. Specifically, the feature-selection-based PCA algorithm applied data pre-processing techniques to reduce the dimension of the RSS measurements of the offline fingerprints database based on their contribution to the positioning performance. In other words, features with a higher explainability variance ratio are the more significant features that discriminate the model positioning performance and should be retained in the model for further analysis. The learned model is then applied during the testing phase after the PCA data preprocessing is performed to infer the mobile user’s location based on the received new RSS measurements by mapping into the highest likelihood resemblance of signal-signature stored in the radio map. The feature selection-based PCA algorithm primarily deals with the dimensionality curse of a predefined radio map by linearly combining the features into an uncorrelated space (i.e., eigenspace) using the predefined radio map’s training covariance matrix of size $F^{\prime }\times F^{\prime }$. The fingerprint database is then projected into uncorrelated space in the direction of the highest variance ratio and the principal components are chosen based on the highest explainability variance ratio (i.e., eigenvalue) which represents the information context of the principal components. Finally, Algorithm 1 below also includes the detailed steps for the PCA.
**Algorithm 1.** Construction of the new fingerprint feature vectors based PCA1. **Input:** Database fingerprint $F_{ij}^{\left(g\right)}=\left\{F_{ij}^{\left(g\right)};\;\left(x_{g},y_{g}\right)\right\}=\left\{S_{ij}^{\left(g\right)}\right\};\;g=0,\;1,\;\ldots ,\;k-1;i=1,\;2,\;\ldots ,\;n\;and\;j=1,\;2,\;\ldots ,\;m$
2. **Input:** Received RSS testing data $F_{ij}^{\left(g\right)}=\left\{S_{ij}^{\left(g\right)};\;\left(?,\;?\right)\right\};\;g=0,\;1,\;\ldots ,\;k-1;\;i=1,\;2,\;\ldots ,\;n_{t}\;and\;j=1,\;2,\;\ldots ,\;m_{t}$
3. **Output:** Source fingerprint $F_{ij}^{\prime }\left(g\right)=\left\{F_{ij}^{\prime }\left(g\right);\;\left(x_{g},\;y_{g}\right)\right\}=\left\{s{s^{\prime }}_{ij}^{\left(g\right)}\right\};\;i=1,\;2,\;\ldots ,\;n^{\prime }\;and\;j=1,\;2,\;\ldots ,\;m^{\prime }$
4. **for** g=0:k−1 do5.          **for** j=0:m−1 do6.               **for** i=0:n−1 do7.                          From each measurement subtract average value: $S_{ij}^{\left(g\right)}\leftarrow S_{ij}^{\left(g\right)}-\frac{1}{n}\sum_{i=1}^{n}S_{ij}^{\left(g\right)};$
8.                          Find the covariance matrix of the measurements: $\sum_{i=1}^{n}\left(S_{i}-{\bar{S}}_{i}\right){\left(S_{i}-{\bar{S}}_{i}\right)}^{T}=SS^{T};$
9.                          Calculate eigenvectors and eigenvalues of the $SS^{T}$;10.                        Create feature vectors $q_{1},\;q_{2},\;\ldots ,\;q_{F^{\prime }}$ corresponding to the largest ${F^{\prime }}_{ij}^{\left(g\right)}$ eigenvalues.11.                        Obtain projection matrix: $Q^{*}=\left(q_{1},\;q_{2},\;\ldots ,\;q_{F^{\prime }}\right)$
12.               **end for**13.         **end for**14. **end for**15. return $F_{ij}^{\prime }$


### 3.2. Heterogeneous Transfer Learning

Suppose the Wi-Fi RSS-based fingerprint positioning was conducted in an indoor environment setting and the fingerprint database is given as: Fij(g)={Fij(g);(xg,yg)};i=1, 2, …, ns & j=1, 2, …, ms also known as the labeled source data. Whereas the (xk,yk) denotes the corresponding coordinate to the associated location of the signal signature and the target instances would be received during the testing phase as Fij(g)={Fij(g)};i=1, 2, …, nt & j=1, 2, …, mt. Similarly, nt and ns represent the numbers of instances for the target and source domains, respectively. We have noted that there are two major challenges that need to be considered in heterogenous transfer learning: (a) The Wi-Fi signal strength received at a grid point from multiple APs assumed to be independent such that the Wi-Fi signals transmitted from different APs are transmitted independently and do not interfere with each other. Nonetheless, the RSS values of a grid point may be duplicated possibly from multiple Wi-Fi APs and creates interference of matched patterns with the actual user’s fingerprint or irrelevant features might be generated and recorded in the database which could reduce our positioning accuracy. (b) Dimensions of the source feature spaces and target domains may also be different (i.e.,ℝFs≠ℝFt). In this paper, the noise created due to the duplicated fingerprints and interdependence of APs are handled by using both the PCA (as in Algorithm 1) and correlation coefficient techniques which could decrease the dependency of certain Wi-Fi APs or extract the most significant fingerprint features that could be used to build the homogeneous feature spaces. The RSS values of the source domain can be given as: F′S(g′)={F′i′j′(g′);(x′g′,y′g′)}={ss′ij(g′)}; i′=1, 2, …, n′s & j′=1, 2, …, m′s and the target domain is represented as: F′T(g)={F′i′j′(g);(x′g′)}; i′=1, 2, …, n′t & j′=1, 2, …, m′t. The Wi-Fi signals received at a grid point from multiple independent APs can be defined as: Sij(g)=∪i=1nSj(g);   j=0, 1, …, m−1,  i=1, …, n and P(∩j=0m−1Sij(g))=P(∪i=1ns0(g)).P(∪i=1ns1(g))….P(∪i=1nsm−1(g)). Note that the *i*th
 Wi-Fi signal strength of fingerprints received at the *k*th grid point of the *j*th Wi-Fi AP is a vector and the fingerprint database can be represented as a matrix Fij(g)=[S1, S2, …, Sm] where Sj=[s00, s01, …, s0n]T. The correlation coefficient of the RSS measurements of the Wi-Fi access points can be determined as:(4)r=cov(Sj,Sj+1)sj2sj+12
where cov(Sj,Sj+1) denotes the covariance of Sj and Sj+1 which is given as follows:(5)cov(Sj,Sj+1)=∑(Sj−S¯j)(Sj+1−S¯j+1)n−1

$s_{j}^{2}$ and $s_{j+1}^{2}$ represent the variances of the measurements of $S_{j}$ and $S_{j+1}$. The pseudo code for positioning using heterogenous TL with the hybrid-based feature selection approach is provided in Algorithm 2.
**Algorithm 2.** The proposed heterogenous transfer learning for Wi-Fi indoor positioning hybrid-based1. **Input:** Source fingerprint $F_{ij}^{\prime }\left(g\right)=\left\{F_{ij}^{\prime }\left(g\right);\;\left(x_{g},y_{g}\right)\right\}=\left\{{{\mathrm{ss}}^{\prime }}_{ij}^{\left(g\right)}\right\};\;i=1,\;2,\;\ldots ,\;n^{\prime }\;and\;j=1,\;2,\;\ldots ,\;m^{\prime }$2. **Input:** Received RSS testing data $F_{T}^{\left(g\right)}=\left\{F_{i^{\prime }j^{\prime }}^{\left(g\right)};\;\left({x^{\prime \prime }}_{g^{\prime }},?\right)\right\};\;g^{\prime }=0,1,\ldots ,k^{\prime }-1;\;i^{\prime }=1,2,\ldots ,{n^{\prime }}_{t}\;\&\;j^{\prime }=1,2,\ldots ,{m^{\prime }}_{t}$3. **Output:** Domain mapping of ${F^{\prime }}_{S}^{\left(g^{\prime }\right)}$,${F^{\prime }}_{T}^{\left(g^{\prime }\right)}$; transfer weight $\beta_{i^{\prime }j^{\prime }}$; predicted labels $\left({x^{\prime }}_{g^{\prime }},{y^{\prime }}_{g^{\prime }}\right)$;4. **for** $g^{\prime }=0:k^{\prime }-1$ do5.         **for** $j^{\prime }=0:m^{\prime }-1$ do6.               **for** $i^{\prime }=0:n^{\prime }-1$ do7.                      Apply PCA as detailed in Algorithm 18.                      Obtain projection matrix: $Q^{*}=\left(q_{1},q_{2},\ldots ,q_{F^{\prime }}\right)$
9.                      Obtain the correlation coefficient matrix using Equation (4): ${\left(r_{i^{\prime }j^{\prime }}\right)}_{N^{\prime }M^{\prime }}$
10.                    Select uncorrelated predictors or significant predictors for modeling the variance11.                    Compute $\beta_{i^{\prime }j^{\prime }}$ by using Equation (7)12.              **end for**13.         **end for**14. **end for**15.  **until** convergence16. Train a classifier from ${F^{\prime }}_{S}^{\left(g^{\prime }\right)}=\left\{{F^{\prime }}_{i^{\prime }j^{\prime }}^{\left(g^{\prime }\right)};\;\left({x^{\prime }}_{g^{\prime }},{y^{\prime }}_{g^{\prime }}\right)\right\}$ with considering weights of source domains $\beta_{i^{\prime }j^{\prime }}$
17. Estimate mobile user’s location $\left({x^{\prime }}_{g^{\prime }},{y^{\prime }}_{g^{\prime }}\right)$ on $\left\{{F^{\prime }}_{T}^{\left(g^{\prime }\right)}\right\}$ by applying the trained classifier $f\left(\left\{{F^{\prime }}_{S}^{\left(g^{\prime }\right)};{\left({x^{\prime }}_{g^{\prime }},{y^{\prime }}_{g^{\prime }}\right)}_{S}\right\},\beta_{i^{\prime }j^{\prime }}\right)$
18. **return** $\left\{{F^{\prime }}_{S}^{\left(g^{\prime }\right)};{\left({x^{\prime }}_{g^{\prime }},{y^{\prime }}_{g^{\prime }}\right)}_{S}\right\}$,$\left\{{F^{\prime }}_{T}^{\left(g^{\prime }\right)};{\left({x^{\prime }}_{g^{\prime }},{y^{\prime }}_{g^{\prime }}\right)}_{T}\right\}$, $\beta_{i^{\prime }j^{\prime }}$


### 3.3. The Objective Function

Due to the inherent heterogeneity of hardware devices and the time difference between measurements for each grid point, there is no guarantee that the signals at any given position will be represented by a single value even using the same device. Two other challenges must also be addressed in heterogeneous transfer learning: it assumes that the Wi-Fi signals transmitted from different APs are independent so that the Wi-Fi signals transmitted from individual APs do not interfere with each other. Grid points’ RSS fingerprint values may be duplicates from multiple APs, which may interfere with matched patterns with the actual fingerprint of a user, or irrelevant features could be generated and recorded in our database, lowering positioning accuracy. Furthermore, the dimensions of the source feature spaces and target domains may be different. The noise caused by duplicated fingerprints and APs interdependence is handled by combining PCA and correlation coefficient techniques, which allow us to extract the most significant fingerprint features; thus, enabling the construction of a homogeneous independent feature vector. As a result, the objective function was minimized over the new feature vector to leverage the most significant features, independent features, and related source knowledge to the target domain. The Wi-Fi signal strength vectors of the source fingerprints can be given as: ${F^{\prime }}_{S}^{\left(g^{\prime }\right)}=\left\{{F^{\prime }}_{i^{\prime }j^{\prime }}^{\left(g^{\prime }\right)};\left({x^{\prime }}_{g^{\prime }},{y^{\prime }}_{g^{\prime }}\right)\right\}=\left\{{S^{\prime }}_{S}^{\left(g^{\prime }\right)}\right\};i^{\prime }=1,\;2,\;\ldots ,\;{n^{\prime }}_{s}\;\&\;j^{\prime }=1,\;2,\;\ldots ,\;{m^{\prime }}_{s}$ and the target fingerprints also is given as: ${F^{\prime }}_{T}^{\left(g\right)}=\left\{{F^{\prime }}_{i^{\prime }j^{\prime }}^{\left(g\right)}\right\}=\left\{{S^{\prime }}_{T}^{\left(g^{\prime }\right)}\right\};i^{\prime }=1,\;2,\;\ldots ,\;{n^{\prime }}_{t}\;\&\;j^{\prime }=1,\;2,\;\ldots ,\;{m^{\prime }}_{t}.$ The Minkowski is a generalized distance metric between two vectors and defined as:(6)$${\left\|D\right\|}_{p}^{p}={\sum_{i^{\prime }=1}^{n}\left|{S^{\prime }}_{S}^{\left(g^{\prime }\right)}-{S^{\prime }}_{T}^{\left(g^{\prime }\right)}\right|}^{p}$$
where ${\left\|D\right\|}_{1}\;and\;{\left\|D\right\|}_{2}$ denotes Manhattan [77,78] and Euclidean distances [62,63,79], respectively. The transfer coefficients ($\beta_{i^{\prime }j^{\prime }}$) constraint is to better minimize the signal measurements of differences between the samples of the source and target domains. The equality constraint mentioned in the objective function would assign higher weights to the most related source samples and the lesser weights would be assigned to the less related source samples. The weights are therefore updated as follows. The new feature vectors were used to minimize the received signal strength difference, such that the transfer coefficients can be estimated as:(7)$$\begin{array}{l} \min\sum_{i^{\prime }=1}^{{n^{\prime }}_{S}}\sum_{j^{\prime }=1}^{{n^{\prime }}_{t}}\beta_{i^{\prime }j^{\prime }}\left\|{S^{\prime }}_{S}^{\left(g^{\prime }\right)}-{S^{\prime }}_{T}^{\left(g^{\prime }\right)}\right\|\\ s.t.\sum_{i^{\prime }=1}^{{n^{\prime }}_{S}}e^{-\beta_{i^{\prime }j^{\prime }}}=1,\;j\prime =1,\;2,\;\ldots ,\;{n^{\prime }}_{t} \end{array}$$

The Lagrangian multiplier method was used to solve the constrained optimization problem of Equation (7) and we assumed that we obtained a location estimate $\beta_{{}_{i^{\prime }j^{\prime }}}^{\left(t-1\right)}$ at the (*t* − 1)th iteration, and now we need to estimate the coefficients $\beta_{{}_{i^{\prime }j^{\prime }}}^{\left(t\right)}$ at the *t*th iteration. By applying the Lagrangian multiplier method, we obtain:(8)$$L\left(\beta_{ij},\delta \right)=\sum_{i^{\prime }=1}^{n^{\prime }{}_{S}}\sum_{j^{\prime }=1}^{n^{\prime }{}_{t}}\beta_{i^{\prime }j^{\prime }}\left\|{S^{\prime }}_{S}^{\left(g^{\prime }\right)}-{S^{\prime }}_{T}^{\left(g^{\prime }\right)}\right\|+\delta \left(\sum_{i^{\prime }=1}^{n_{S}}e^{-\beta_{i^{\prime }j^{\prime }}}-1\right)=0$$
where δ is the Lagrangian multiplier. By letting the partial derivative of the Lagrangian with respect to $\beta_{i^{\prime }j^{\prime }}$ and δ to be zeros, we obtain:(9)$$\frac{\partial }{\partial \beta_{i^{\prime }j^{\prime }}}L\left(\beta_{j^{\prime }},\delta \right)=\left\{ \begin{array}{l} \left\|{S^{\prime }}_{S}^{\left(g^{\prime }\right)}-{S^{\prime }}_{T}^{\left(g^{\prime }\right)}\right\|-\delta e^{-\beta_{1j^{\prime }}}=0\\ \left\|{S^{\prime }}_{S}^{\left(g^{\prime }\right)}-{S^{\prime }}_{T}^{\left(g^{\prime }\right)}\right\|-\delta e^{-\beta_{2j^{\prime }}}=0\\ ,\\ \left\|{S^{\prime }}_{S}^{\left(g^{\prime }\right)}-{S^{\prime }}_{T}^{\left(g^{\prime }\right)}\right\|-\delta e^{-\beta_{n_{s}j^{\prime }}}=0\\ \sum_{i=1}^{n_{s}}e^{-\beta_{i^{\prime }j^{\prime }}}-1=0 \end{array}\right.$$

By adding up the first $n_{s}$ terms in Equation (9), we can obtain:(10)$$\delta =\sum_{i^{\prime }=1}^{n_{S}}\left\|{S^{\prime }}_{S}^{\left(g^{\prime }\right)}-{S^{\prime }}_{T}^{\left(g^{\prime }\right)}\right\|$$

Additionally, substituting Equation (10) into Equation (9) gives the estimated transfer coefficients as:(11)$$\beta_{i^{\prime }j^{\prime }}=-\ln\left(\left\|{S^{\prime }}_{S}^{\left(g^{\prime }\right)}-{S^{\prime }}_{T}^{\left(g^{\prime }\right)}\right\|\right)+\ln\left(\sum_{i^{\prime }=1}^{n_{s}}\left\|{S^{\prime }}_{S}^{\left(g^{\prime }\right)}-{S^{\prime }}_{T}^{\left(g^{\prime }\right)}\right\|\right)$$

### 3.4. Prediction Evaluation Metrics

In this paper, we compared the performance of the proposed algorithms against different machine learning algorithms taken as baselines and verified through extensive experimentations and simulations using a real-world dataset. The mean absolute error (MAE) is defined as the average of all absolute errors and was used to evaluate the effectiveness of the proposed algorithm and given as:(12)$$MAE=\frac{1}{n_{t}}\sum_{i^{\prime }=1}^{n_{t}}\left[\left|x_{i^{\prime }}-{\hat{x}}_{i^{\prime }}\right|+\left|y_{i^{\prime }}-{\hat{y}}_{i^{\prime }}\right|\right]$$
where ${\left[{\hat{x}}_{i^{\prime }},{\hat{y}}_{i^{\prime }}\right]}^{T}$ and ${\left[x_{i^{\prime }},y_{i^{\prime }}\right]}^{T}$ are the predicted location estimate and the true location of the mobile user, respectively. Additionally, *n* is the total number of samples to be located in the target domain.

### 3.5. Analysis of Indoor Positioning Performance-Based Cramer–Rao Lower Bound (CRLB)

Recall the general RSS fingerprint-based positioning of the indoor environment scenario partitioned into G grid points (GPs) such that $G=\left\{gp_{g};\left(x_{g},y_{g}\right)|g=0,\;1,\;\ldots ,\;k-1\right\}$ defines the set of reference points associated to the coordinates representing the mobile user’s locations nearest by and each GP is indexed with a label g, and there were M detectable Wi-Fi APs (j). Thus, the *i*th Wi-Fi signal strength of fingerprints received at the *k*th grid point of the *j*th access point is a vector and the fingerprint database from all Wi-Fi APs entirely can be represented as a matrix: $F_{ij}^{\left(g\right)}=\left\{F_{ij}^{\left(g\right)};\left(x_{g},y_{g}\right)\right\}=\left\{ss_{ij}^{\left(g\right)}\right\};i=1,\;2,\;\ldots ,\;n\;\&\;j=1,\;2,\;\ldots ,\;m$ and

$\left(x_{g},y_{g}\right)$ is the corresponding coordinate to the associated location of the signal signature. Additionally, the target instances of signal strengths received during the testing phase can be represented as: ${\left\{F_{ij}^{\left(g\right)}\right\}}_{T}={\left\{ss_{ij}^{\left(k\right)}\right\}}_{T};i=1,\;2,\;\ldots ,\;n_{t}\;\&\;j=1,\;2,\;\ldots ,\;m_{t}$. The main goal of indoor positioning is to improve positioning performance mainly for better location-based services. In line with this, we applied the Cramer–Rao lower bound (CRLB) as it is widely used to evaluate the performance of the indoor positioning algorithm [79] and estimates a lower limit for the variance of any unbiased estimator of an unknown parameter, which is widely used to analyze the localization performance [80,81,82,83]. Thus, the Wi-Fi received signal strengths were used to analyze the lower bound of the location estimation error and it is significantly important to characterize the properties of this lower bound in order to evaluate the impact of different parameters on the accuracy of target positioning. Furthermore, the lower bound analysis can also provide important system design suggestions by revealing error trends with the indoor localization system deployment. Suppose L is an unknown deterministic parameter which is to be estimated from n independent observations (measurements) of $s_{ij}^{\left(g\right)}$, each from a distribution according to some probability density function $f\left(s_{ij}^{\left(g\right)};L\right)$. The variance of any unbiased estimator $\hat{L}$ of L is then bounded by the reciprocal of the Fisher information J(L): $\mathrm{var}\left(\hat{L}\right)\geq {\left[J\left(L\right)\right]}^{-1}$. Where the Fisher information J(L) is defined as:(13)$$J\left(L\right)=nE_{L}\left[{\left(\frac{\partial l\left(S;L\right)}{\partial L}\right)}^{2}\right]$$
where $l\left(s_{ij}^{\left(g\right)};L\right)=\log\left(f\left(s_{ij}^{\left(g\right)};L\right)\right)$ is the natural logarithm of the likelihood function for a single sample $s_{ij}^{\left(g\right)}$ and $E_{L}$ denotes the expected value with respect to the density $f\left(s_{ij}^{\left(g\right)};L\right)$ of S. If $l\left(s_{ij}^{\left(g\right)};L\right)$ is twice differentiable and holds certain regularity conditions, then the Fisher information can also be defined as:(14)$$J\left(L\right)=-nE_{L}\left(\frac{\partial^{2}l\left(S;L\right)}{\partial L^{2}}\right)$$

The efficiency of an unbiased estimator $\hat{L}$ measures how close this estimator’s variance comes to this lower bound; and efficiency of an estimator is defined as
(15)$$e\left(\hat{L}\right)=\frac{{\left[J\left(L\right)\right]}^{-1}}{\mathrm{var}\left(\hat{L}\right)}$$

Along with this, consider that the unknown position of the mobile user and the position of the *j*th access point are denoted as $L={\left(x^{\prime },y^{\prime }\right)}^{T}$ and $L_{j}={\left({x^{\prime }}_{j},{y^{\prime }}_{j}\right)}^{T}$, respectively, then the distance between the mobile user and the *j*th Wi-Fi access point can be calculated as $d_{0;j}={\left[{\left(L-L_{j}\right)}^{T}\left(L-L_{j}\right)\right]}^{\frac{1}{2}}$. Now, the Wi-Fi received signal strengths of a grid point from multiple access points is expected to have fluctuating signal values due to the complex nature of indoor environment possibly affected by the path loss, shadowing, and multipath effect propagation. Hence, the distribution of those measurements could be affected by a random heterogeneity over time such that it is paramount to know how to estimate the random effect of that variable on the overall process of the system modeling. In this case, we adopted the assumption that the random error or the noise could be introduced due to the random phenomenon following a normal distribution with mean zero and variance $\sigma^{2}$ and hence the RSS values can be modeled as:(16)$$SS_{ij}^{\left(k\right)}=L+\xi_{ij}^{\left(k\right)}$$
where g=0, 1, …, k−1;i=1, 2, …, n & j=1, 2, …, m and $\xi_{ij}^{\left(k\right)}∼N\left(0,\sigma_{\varepsilon }{}^{2}\right)$. Additionally, the pdf (probability density function) of the measurements $SS_{ij}^{\left(k\right)}$ is given by:(17)$$f\left(ss_{ij}^{\left(g\right)};L\right)=\frac{1}{{\left(2\pi \sigma_{\varepsilon }{}^{2}\right)}^{1/2}}e^{\left(\frac{-1}{2\sigma_{\varepsilon }{}^{2}}{\left(ss_{ij}^{\left(g\right)}-L\right)}^{2}\right)}$$

In RSS fingerprint-based indoor localization, we assume that the Wi-Fi received signal strengths from multiple APs at a reference point are independent and identically distributed such that the joint likelihood function for the measurements can be defined as:(18)$$f\left(ss_{ij}^{\left(g\right)};L\right)=\prod_{j=0}^{M-1}\frac{1}{{\left(2\pi \sigma_{\varepsilon }{}^{2}\right)}^{1/2}}e^{\left(\frac{-1}{2\sigma_{\varepsilon }{}^{2}}{\left(ss_{ij}^{\left(g\right)}-L\right)}^{2}\right)}$$

By exponential property, Equation (18) can be rewritten as:(19)$$f\left(ss_{ij}^{\left(g\right)};L\right)=\frac{1}{{\left(2\pi \sigma_{\varepsilon }{}^{2}\right)}^{N/2}}e^{\left(\frac{-1}{2\sigma_{\varepsilon }{}^{2}}\sum_{g=0}^{K-1}\sum_{j=0}^{M-1}\sum_{i=0}^{n-1}{\left(ss_{ij}^{\left(g\right)}-L\right)}^{2}\right)}$$

Take ln to get the log-likelihood function of the above Equation (19):(20)$$\ln f\left(ss_{ij}^{\left(g\right)};L\right)=-\ln\left[{\left(2\pi \sigma_{\varepsilon }{}^{2}\right)}^{N/2}\right]-\left(\frac{1}{2\sigma_{\varepsilon }{}^{2}}\sum_{g=0}^{K-1}\sum_{j=0}^{M-1}\sum_{i=0}^{n-1}{\left(ss_{ij}^{\left(g\right)}-L\right)}^{2}\right)$$

Apply the first partial derivative w.r.t L:(21)$$\frac{\partial }{\partial L}\ln f\left(ss_{ij}^{\left(g\right)};L\right)=\frac{1}{\sigma_{\varepsilon }{}^{2}}\sum_{g=0}^{K-1}\sum_{j=0}^{M-1}\sum_{i=0}^{n-1}\left(ss_{ij}^{\left(g\right)}-L\right)=\frac{KMn}{\sigma_{\varepsilon }{}^{2}}\left(s{\bar{s}}_{ij}^{\left(g\right)}-L\right)$$

Take the second partial derivative of equation:(22)$$\frac{\partial^{2}}{\partial L^{2}}\ln f\left(ss_{ij}^{\left(g\right)};L\right)=-\frac{KMn}{\sigma_{\varepsilon }{}^{2}}$$

Additionally, one can see that the final result does not depend on the vector of observations $S_{ij}^{\left(g\right)}$ and the CRLB of the estimate is given as:(23)$$\sigma^{2}\left(\hat{L}\right)\geq \frac{1}{J\left(L\right)}=\frac{1}{-E\left[\frac{\partial^{2}}{\partial L^{2}}\ln f\left(ss_{ij}^{\left(g\right)};L\right)\right]}=\frac{\sigma_{\varepsilon }{}^{2}}{KMn}$$

We proved that the variance of the location estimator $\left(\hat{L}\right)$ attained the lower boundary variance (CRLB) and this indicates our estimator is also the MVU (Minimum Variance Unbiased) estimator as no unbiased estimator could do better and the Fisher information matrix (FIM) [82,83] is given by
(24)$$J\left(L\right)=nE_{L}\left[{\left(\frac{\partial l\left(S_{{}_{ij}}^{\left(g\right)};L\right)}{\partial L}\right)}^{2}\right]$$

If l(x;L) is twice differentiable and holds certain regularity conditions, then the Fisher information can also be rewritten as:(25)$$J\left(L\right)=-nE_{J}\left(\frac{\partial^{2}l\left(S_{{}_{ij}}^{\left(g\right)};L\right)}{\partial L^{2}}\right)$$

Therefore, the CRLB of MSE of the L can be calculated as
(26)$$CRLB=\frac{J_{xx_{i}}\left(L\right)+J_{yy_{i}}\left(L\right)}{J_{xx_{i}}\left(L\right).J_{yy_{i}}\left(L\right)-J_{xy_{i}}\left(L\right)J_{yx_{i}}\left(L\right)}$$

From Equations (19), (24), and (25), we see that:(27)$$J\left(L\right)=nE_{L}\left[{\left(\frac{\partial l\left(ss_{ij}^{\left(g\right)};L\right)}{\partial L}\right)}^{2}\right]=nE_{L}\left[{\left(\frac{\partial }{\partial L}\ln f\left(ss_{ij}^{\left(g\right)};L\right)\right)}^{2}\right]=nE_{L}\left[{\left[\frac{1}{\sigma_{\varepsilon }{}^{2}}\sum_{g=0}^{K-1}\sum_{j=0}^{M-1}\sum_{i=0}^{n-1}\left(ss_{ij}^{\left(g\right)}-L\right)\right]}^{2}\right]$$
(28)$$=nE_{L}\left[\frac{1}{\sigma^{4}}\left[\sum_{g=0}^{K-1}\sum_{j=0}^{M-1}\sum_{i=0}^{n-1}{\left(\left(x-x_{i}\right)-\left(y-y_{i}\right)\right)}^{2}\right]\right]$$
(29)$$J_{xx_{i}}\left(L\right)=\frac{n}{\sigma_{\xi }^{4}}\sum_{g=0}^{K-1}\sum_{j=1}^{M-1}\sum_{i=0}^{n-1}E_{L}\left({\left[\left(x-x_{i}\right)\right]}^{2}\right)$$
(30)$$J_{yy_{i}}\left(L\right)=\frac{n}{\sigma_{\xi }^{4}}\sum_{g=0}^{K-1}\sum_{j=0}^{M-1}\sum_{i=0}^{n-1}E_{L}{\left[\left(y-y_{i}\right)\right]}^{2}$$
(31)$$J_{xy_{i}}\left(L\right)=\frac{n}{\sigma_{\xi }^{4}}\sum_{g=0}^{K-1}\sum_{j=0}^{M-1}\sum_{i=0}^{n-1}E_{L}\left[\left(x-x_{i}\right)\left(y-y_{i}\right)\right]$$

On the other hand, the CRLB analysis can be restated using the concept of mean square error (MSE). Assume L is an unknown deterministic parameter which is to be estimated from n independent observations (measurements) of $ss_{ij}^{\left(g\right)}$, each from a distribution according to some probability density function $f\left(ss_{ij}^{\left(g\right)};L\right)$. The MSE of any estimator $\hat{L}$ of L is then defined as:(32)$$MSE\left(\hat{L}\right)=E\left[{\left(\hat{L}-L\right)}^{2}\right]=E\left\{{\left[\left(\hat{L}-E\left(\hat{L}\right)\right)+\left(E\left(\hat{L}\right)-L\right)\right]}^{2}\right\}$$
(33)$$=\mathrm{var}\left(\hat{L}\right)+{\left[E\left(\hat{L}\right)-L\right]}^{2}+2E\left\{\left[E\left(\hat{L}\right)-\hat{L}\right]\left[E\left(\hat{L}\right)-L\right]\right\}$$

From properties of expectations of random variables, the last term of Equation (33) would give zero. Additionally, the term $\mathrm{var}\left(\hat{L}\right)$ represents the variance of the estimator $\hat{L}$. Finally, as in Equation (33), the MSE is defined as the sum of two components of the variance of the estimator ($\mathrm{var}\left(\hat{L}\right)$) and the square of the differences between the expectation of the estimator and the actual mobile user’s location $\left({\left[E\left(\hat{L}\right)-L\right]}^{2}\right)$. Alternatively, for the unbiased estimator of $\hat{L}$, its $MSE\left(\hat{L}\right)$ must be equal to $\mathrm{var}\left(\hat{L}\right)$ and consequently the unbiased estimator of L, i.e., $E\left(\hat{L}\right)=L$ and to the right side of Equation (33) the second sum term would vanish for large samples. Moreover, let us recall the above scenario of the RSS fingerprint-based indoor positioning system such that $\hat{L}=\left(\hat{x^{\prime }},\hat{y^{\prime }}\right)$ is the estimate of the unknown mobile user’s coordinates $L=\left(x^{\prime },y^{\prime }\right)$, and the covariance matrix of the estimator can be given as:(34)$$Cov\left(\hat{L}\right)=E\left\{\left(\hat{L}-L\right){\left(\hat{L}-L\right)}^{T}\right\}=\left[\begin{array}{cc} E{\left(\hat{x^{\prime }}-x^{\prime }\right)}^{2} & E\left[\left({\hat{y}}^{\prime }-y^{\prime }\right)\left(\hat{x^{\prime }}-x^{\prime }\right)\right]\\ E\left[\left(\hat{x^{\prime }}-x^{\prime }\right)\left({\hat{y}}^{\prime }-y^{\prime }\right)\right] & E{\left({\hat{y}}^{\prime }-y^{\prime }\right)}^{2} \end{array}\right]\geq {\left[J\left(L\right)\right]}^{-1}$$
where $E\left[\left(\hat{x^{\prime }}-x^{\prime }\right)\left({\hat{y}}^{\prime }-y^{\prime }\right)\right]$ is the covariance between ${\hat{x}}^{\prime }$ and ${\hat{y}}^{\prime }$ which can be represented as $\mathrm{cov}\left({\hat{x}}_{i}^{\prime },{{\hat{y}}^{\prime }}_{i}\right)$ and since the covariance is commutative it is equal with $\mathrm{cov}\left({\hat{y}}_{i}^{\prime },{{\hat{x}}^{\prime }}_{i}\right)$. The term J(L) refers to the Fisher information matrix (FIM) and Equation (28) holds the criterion of CRLB. Moreover, if $f\left(S_{ij}^{\left(g\right)};L\right)$ denotes the likelihood function of measurements $S_{ij}^{\left(g\right)}$ conditioned on L, then the score function is defined as the gradient of its log-likelihood function such that:(35)$$\lambda \left(L\right)=\nabla \ln f\left(S_{ij}^{\left(g\right)};L\right)=\frac{\partial \ln f\left(S_{ij}^{\left(g\right)};L\right)}{\partial L}$$

Additionally, the FIM can be defined as the variance of this score function λ(L):(36)$$\sigma^{2}\left(\lambda \left[L\right]\right)=E\left[{\left(\frac{\partial \ln f\left(S_{ij}^{\left(g\right)};L\right)}{\partial L}\right)}^{2}\right]=J\left(L\right)$$

One can also observe that
(37)E[λ[L]]=0

From the above derivation, we noted that an interesting relationship for the FIM above and the signal model we considered in Equation (16) were revealed as a reciprocal multiplicative effect to the Gaussian random noise that might be introduced due to the heterogeneity nature of complex indoor environment settings and can be described as: (38)$$J_{xx_{i}}\left(L\right)=\frac{n}{\sigma_{\xi }^{4}}\sum_{g=0}^{K-1}\sum_{j=1}^{M-1}\sum_{i=0}^{n-1}E_{L}\left({\left[\left(x-x_{i}\right)\right]}^{2}\right).$$

## 4. Experimental Results and Discussion

In this section, we present a real-world experiment to evaluate the proposed algorithms. Experimental settings and dataset are presented first, followed by an analysis of the overall performance of the classifiers.

### 4.1. Experimental Settings

#### Dataset

An experiment with an area of 1460 m^2^ and 175 reference points (RPs) equidistant from the adjacent next grid point was carried at the University of Electronic Science and Technology of China (UESTC) on the 21st floor of the innovation building as shown on Figure 2. There were nine Wi-Fi access points (Wi-Fi APs) available to collect the received signal strength from a mobile device, allowing for the creation of the fingerprint database, and they were sparsely deployed to ensure that at least three Wi-Fi APs could be detected at each grid point. The layout and experimental settings of the generated real office experiment scenario for the dataset are shown below.

### 4.2. Distribution of the RSS Measurements

Figure 3 depicts the distribution of the new feature vector along with the labels for both training and testing Wi-Fi RSS values for the dataset. Despite a slight mix-up of fingerprints across the labels due to the heterogeneous nature of signal power attenuation over distance, the RSS measurements received from multiple base stations at a grid point appear to have shown a specific distribution. As shown in Figure 3, the two main principal components referred to as the ‘base model’ are considered to be the simplest model and enable visualization of the effect of variations for mobile user positioning by retaining only the most significant features.

A ‘base model’ is one with only two major principal components that represent the variational distributions of the target’s prediction and account for approximately 56% of the model’s total variance explainability. From a technical standpoint, this has two major advantages: computational cost and model simplicity may improve as feature space dimensions are reduced. Furthermore, larger feature spaces would necessitate the deployment of more base stations, which would be prohibitively expensive in practice, and redundant features could introduce systematic errors into the entire positioning system modeling process. Because there is a tradeoff over accuracy due to the removal of some features or a reduction in dimension of feature spaces, there are some practical situations where the major principal components explaining only 56% of the model’s prediction are a better choice. It is also pertinent to note that training and testing data may have different distributions.

Figure 4 depicts the distribution of principal components accounting for 95 percent explained variance ratio (EVR) for the dataset, along with their corresponding labels. Figure 4a,b show the distributions of the first and second principal components, which account for approximately 56% and 55% of the total variations that could account for both instances of training and testing datasets, respectively. Even though the two distributions of the base model appear to be different, the RSS measurements received at a reference point from various Wi-Fi access points appear to be independent, and the distribution of each label can be characterized by using the base model, which has only two principal components. This could also demonstrate that keeping the most important predictors could help the indoor positioning system work more efficiently and cost-effectively. Nonetheless, with only two principal components, this basic model could perform mobile user positioning at the expense of some information discarded based on their contributions.

Consider Figure 4c,d to better understand the effect of each principal component’s contribution, which depicts the distributions of the second and third principal components, causing the RSS values to become muddled. This means that the second and third principal components (the second and third) account for approximately 26% and 30% of the total variance explainability ratio for both instances of training and testing datasets, respectively, and appear to be poor fitting in estimating the location of the mobile user when compared to the base model, which accounts for 56% of both domains. This trend is visible in Figure 4e,f, which depicts the distributions of the third and fourth principal components (21 percent and 18 percent for training and testing datasets, respectively), indicating that the RSS values received from multiple base stations at a grid point failed to characterize the ‘signal to location’ relationship or that there is a high tendency of mix-up among the clusters of RSS measurements from a single reference point. The less explained variance ratio by a principal component, the less to characterize the ‘signal–location’ relationship, and the algorithms may fail to estimate the mobile user’s position or degrade the system model’s performance.

The effect of different feature dimension possibilities on the amount of variance ratio explainability by algorithms for mobile user positioning estimation is shown in Table 1. To estimate the location of a mobile user with a 95 percent explained variance ratio, a classifier, for example, requires seven principal components. A classifier, on the other hand, can localize a mobile user using only two main principal components, which account for approximately 56% of the model’s total variations (base model). Various scenarios of algorithms or model-building processes can be selected based on various attributes such as cost, simplicity, and interpretability, as well as algorithm accuracy. The fewer principal components used in a model indicate the lower the computational cost and memory usage. It also includes a tip about how simpler models require fewer feature spaces to properly visualize the model. In real-life scenarios, a model with higher dimensional feature spaces or principal components would be more difficult to understand and interpret. Technically, the more complex the model, the more vulnerable it is to model overfitting as it attempts to represent the highest variance explained ratio.

Table 2 describes the number of measurements collected from each grid point and demonstrates that the label distribution for the dataset is balanced. As a result, feature scaling techniques were not taken into account in order to avoid the dominance effect of the cluster’s higher label occurrence. Otherwise, the cluster’s larger features would dominate the others. To use the principal component analysis, however, features must be standardized, so all values in the dataset are set to be standardized.

Figure 5 depicts a scree plot of the dataset’s 95 percent and 90 percent variance explained ratios and the amount of variance ratio explainability by the principal components for mobile user positioning accuracy. The y- and x-axes represent the eigenvalues and the number of principal components, respectively. For both instances of training and testing datasets, the scree plot shows a sharp decrease from principal component 1 to principal component 2, and then a slow decremental value or slope from principal component 2 through the last principal components. Furthermore, the general rule of thumb for interpreting the scree plot is to keep the number of principal components above the scree or in areas where the plot does not drop significantly. As a result, the scree plot suggests that we keep only one principal component, which accounts for roughly 47 percent of total model variations. This confirms the significance of principal component one as a predictor of location estimation accuracy, which accounts for approximately 47 percent of the variance explained ratio. Moreover, this has a direct impact on implementation costs, as the deployment of more Wi-Fi access points with less contribution may be due to their receiver’s antenna orientation or other features that are inefficient for the positioning task.

Figure 6a,b further elucidates that the distributions of the sixth and seventh principal components are such that the RSS values are highly mixed-up among the different labels of mobile users in both instances of training and testing phases, or there is no clear demarcation between the clusters of fingerprints generated at each reference point, causing the modeling system to be highly fluctuated due to possible fingerprint duplications at a grid point. This indicates that the two principal components (the sixth and seventh) account for about 9 percent and 11 percent of the total variance explainability ratio for both instances of training and testing datasets, respectively, and appear to be very poor fitting in estimating the location of the mobile user when compared to the base model, which accounts for 56 percent of both domains. The lesser the amount of explained variance ratio by a principal component, the less likely the algorithms are to establish the ‘signal to location’ relationship, and the algorithms may fail to estimate the mobile user’s position or degrade the system model’s performance.

Table 3 depicts the effect of various feature dimension options on the amount of variance ratio explainability by classifiers for mobile user positioning accuracy, taking into account the nature of distributions of instances from both training and testing data points separately. Seven principal components explained 95 percent of the total variations of the model in the training dataset, whereas eight principal components are required to fit the model for testing data points with the same size of variations. Similarly, the 90 percent explained variance ratio required six principal components for training phases and seven principal components for testing phases. The difference in principal component requirements for training and testing datasets appears insignificant for small-sized feature spaces. Consideration of a different number of principal components for the same amount of explained variance ratio for both training and testing datasets, on the other hand, can result in a significant difference for large real-world datasets with potentially large feature spaces, particularly in heterogeneous transfer learning scenarios. This effect of variation in the number of principal components across both domains of training and testing datasets could be demonstrated more clearly for large datasets with large feature spaces.

Table 4 describes the effect of each principal component on the predictive model’s explained variance account for dataset A of both training and testing datasets. As a result, principal component 1 (PC1) contributed the highest explained variance ratio (EVR) for mobile user positioning estimation, followed by principal component 2 (PC2), which contributed the second highest variance account for both training and testing datasets. Furthermore, starting with principal component 3 (PC3), the contributions of each principal component decreased from 12.49 to 4.67 percent.

According to Figure 7, the contribution of each principal component to model variation for indoor positioning of mobile users applied to both training and testing datasets of A decreased from PC1 to PC8 or PC1 constituted the most information for estimating the location of the mobile users. We discovered that principal component 1 (PC1) contributed the highest explained variance ratio (EVR) for mobile user positioning estimation, revealing 42.71 percent and 35.75 percent for training and testing datapoints, respectively. This implies that a classifier can be used to estimate location with smaller feature space dimensions if the most significant variations are retained, despite the fact that some possible variations are excluded from the model.

Figure 8 shows that the contribution of the number of principal components to model variation for indoor positioning of mobile users applied to the training dataset of A appears to be sharply increasing from the base model to PC7 and sharply decreasing from PC6 to PC4. The base model, which consists of only two principal components, achieved approximately 56% of the explained variance ratio, implying that reducing feature spaces would make our model easier to visualize and understand the analysis. On the other hand, the benefits of principal component analysis, such as simplicity, interpretability, and minimizing computational cost, could not be retained at no cost.

Table 5 depicts the application of feature correlation analysis to investigate the presence of multicollinearity, and it can be seen that three predictors appear to have slightly higher correlation values when compared to the overall correlational trend values, indicating that those three values should be investigated further. Furthermore, the overall distributions of those features have mostly similar RSS values, indicating that no signal was received, as indicated by the value −90. As a result, we decided to remove those three features and rebuild our classifier algorithm to see if positioning performance would suffer. Furthermore, the principal component analysis clearly demonstrates that six principal components can explain 90% of the total variations in the model, which is consistent with feature correlational analysis, which demonstrates that six features are independent. This is further supported by the fact that the Wi-Fi received signal strengths at a grid point from multiple access points should be independent; otherwise, the higher correlated samples in the analysis would degrade our positioning system’s accuracy and cost as we deploy more Wi-Fi access points that could no longer bring any unique features in modeling the variance of the positioning system. In other words, redundant features would have a negative impact on positioning performance, so we decided to exclude from the analysis those features that are either redundant or irrelevant, as well as highly correlated features. In our case, features (X4, X6, and X8) were excluded from the analysis due to their higher correlations with other features.

### 4.3. Comparative Analysis of Methods

In this section, the performance of the proposed algorithms for positioning estimation using Wi-Fi RSS-based fingerprints of indoor positioning was compared to the most popular algorithms in the field of prediction tasks and machine learning, in general, using a real-world experimental dataset. Table 6 also describes how several machine learning algorithms were applied to the dataset to characterize the effect of different principal components (PCs) on the model’s target prediction. Five different scenarios were considered to see if the amount of information used for location estimation could change depending on the contribution of each principal component. The scenarios correspond to the percentages of variations explained by the model or the explained variance ratios for the dataset of 80%, 85%, 90%, 95%, and 56%, with the corresponding number of principal components of 4, 5, 6, 7, and 2, respectively. Furthermore, correlation analysis was used to select features based on their impact on positioning performance. To train and estimate the position of a mobile user given the received signal strengths from multiple Wi-Fi access points, an extensive experimentation-based real office scenario of indoor environment settings for the dataset was performed, and the positioning performance of the different classifiers was evaluated (APs).

In this paper, we proposed two feature selection algorithms based on different scenarios: (a) the proposed algorithm relies solely on principal component analysis (PCA) for feature selection, and (b) the hybrid-based approach considers both PCA and correlation effect analysis. Furthermore, for their prediction tasks, the proposed algorithms were compared to the most popular machine learning algorithms, such as decision tree (DT), K nearest-neighbor (KNN), support vector machine (SVC), logistic regression (LR), random forest (RF), and neural network (MLP). To select the most significant predictors and avoid any negative knowledge transfer, data preprocessing was performed, and irrelevant feature spaces were removed from further analysis based on the two criteria listed above in (a) and (b). The original number of feature spaces for the dataset was 9. However, all feature spaces could not be considered for positioning estimation based on the scenarios presented above for three reasons: (a) some features may be irrelevant, resulting in duplicate fingerprints or pattern fingerprint mismatches, (b) large feature spaces necessitate a massive deployment of Wi-Fi access points, which is prohibitively expensive, and (c) dealing with higher dimension features incur significant computational and memory costs. As a result, we proposed the above-mentioned two scenarios of feature selection mechanisms for use in positioning estimation or estimating a mobile user’s location.

Table 7, on the other hand, shows the use of several machine learning algorithms on the dataset to characterize the effect of various principal components (PCs) on the model’s target prediction using transfer learning. Similarly, the above five scenarios were considered to see if the amount of information used for location estimation could vary depending on the contribution of each principal component after transfer learning was applied. As shown in Table 7, the combined use of principal component analysis and transfer learning significantly improved the positioning performance of the classifiers.

Figure 9 shows that the proposed algorithm-based PCA (before transfer learning) was discovered to be the best algorithm with the lowest mean absolute error for the dataset with different principal components, accounting for 80%, 85%, 90%, 95%, and 56% variance explainability of positioning estimation. A classifier, on the other hand, can localize a mobile user using only two principal components, which account for roughly 56% of the model’s total variations (base model). Thus, various scenarios of algorithms or model building processes can be selected based on various attributes such as cost, simplicity, and interpretability, as well as algorithm accuracy. The fewer principal components used in a model, the lower the computational cost and memory usage. In real-life scenarios, a model with higher dimensional feature spaces or principal components would be more difficult to understand and interpret. Technically, the more complex the model, the more vulnerable it is to model overfitting as it attempts to constitute the highest variance explained ratio of the new measurements received.

However, Figure 10 depicts that the proposed algorithm-based PCA after applying transfer learning (HetTLIPPC based: heterogeneous transfer learning for indoor positioning based PCA) also reveals a significant positioning performance improvement and one can clearly observe that the proposed algorithm has come out to be the best algorithm with the lowest mean absolute error applied to the dataset with different principal components, accounting for 80%, 85%, 90%, 95%, and 56% variance explainability of positioning estimation. Surprisingly, the base model that constitutes approximately 56% of the variance explainability ratio of the model performs as the best in both cases. However, the risk of model overfitting, which is a very common phenomenon, is an imminent threat for indoor positioning due to the dynamic nature of the indoor environment described by NLOS propagation, multipath effects, and random signal noise. Thus, we recommend a model with a 95 percent variance of explainability ratio to represent the most discriminant features possible in a complex indoor environment.

The effect of a heterogeneous transfer-learning-based hybrid feature selection approach that incorporates both PCA and correlation effect analysis is shown in Table 8. As shown in Table 5, the principal component analysis clearly shows that six principal components can explain 90% of the total variations in the model, which is consistent with the feature correlational analysis, which shows that six features are independent. The positioning performance of the classifiers clearly improved after applying transfer learning to the hybrid feature selection approach, as shown in Table 7.

## 5. Conclusions

In this paper, we proposed two novel algorithms for feature selection purposes applied to fingerprint-based IP estimation problems to improve positioning performance in the target domain through heterogeneous TL-based hybrid feature selection. The two scenarios of the algorithms are (a) principal component analysis-based feature selection and (b) the hybrid-based approach for feature selection considering both the combination of PCA and correlation effect analysis. To this end, we created a new feature vector on which the model was trained by retaining only the most significant predictors, and we also determined the most efficient feature dimensions using the hybrid-based feature selection approach. We discovered that not all available feature spaces could be considered for positioning estimation for at least four reasons: (a) some features may be irrelevant and result in fingerprint duplication or pattern fingerprint mismatch; (b) large feature spaces necessitate a massive deployment of Wi-Fi access points, which is prohibitively expensive from a cost standpoint; (c) from a technical standpoint, computational cost and memory usage are quite expensive when dealing with higher dimensions of features; and (d) model over-fitting is also a serious issue for higher dimensions of feature spaces.

The hybrid-based proposed algorithm was found to be the best algorithm with the lowest mean absolute error, confirming that our proposed algorithm improved positioning estimation accuracy through heterogeneous transfer learning. Moreover, not only was positive knowledge transferred from the source domain to the target domain, but the computational cost and memory usage were effectively reduced by retaining the most significant predictors using both principal component and correlation analysis techniques. Analysis of the lower bound for the position estimation error based CRLB, on the other hand, shows that the variance of the random noise ($\sigma_{\xi }^{2}$), the distance between the mobile user and Wi-Fi access points ($d_{0,g}$) or the location of Wi-Fi APs and mobile user, the number of Wi-Fi access points, and the signal propagation parameters can affect position estimation using Wi-Fi RSS-based fingerprinting of indoor positioning. Although the number of Wi-Fi APs can affect the lower bound location estimation error, we saw that identifying the most significant features could help to improve the target’s positioning accuracy. This could be accomplished by combining PCA methods and correlation analysis, both of which are commonly used in prediction tasks to exploit the most important features (or measurements with a high explained variation ratio) and avoid measurement interdependence, respectively.

## Figures and Tables

**Figure 1 sensors-22-05840-f001:**
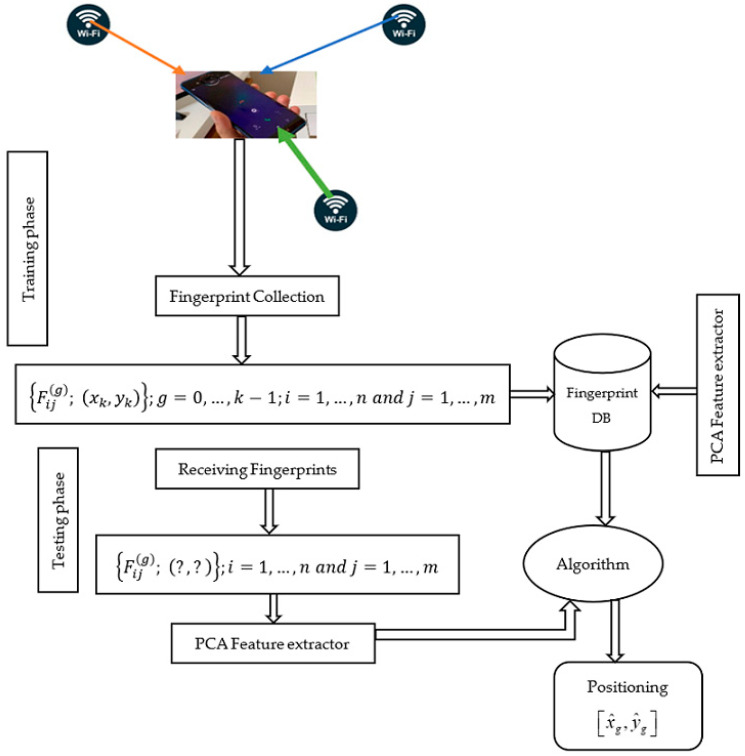
Proposed framework of Wi-Fi received signal strength-based fingerprint for the indoor positioning system.

**Figure 2 sensors-22-05840-f002:**
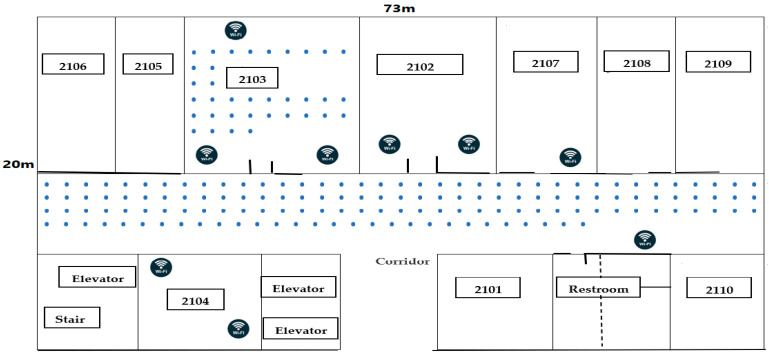
Experimental layout conducted in UESTC for generating the dataset.

**Figure 3 sensors-22-05840-f003:**
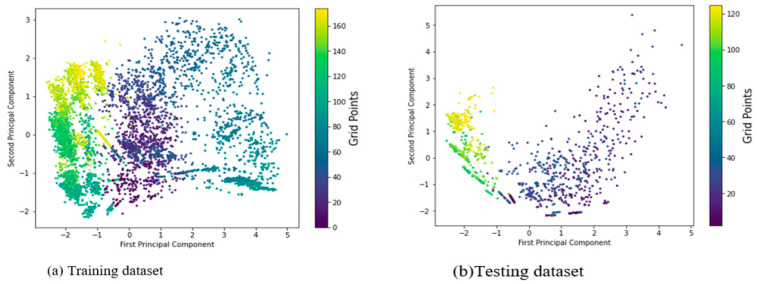
Distribution of principal components with their corresponding labels for the dataset.

**Figure 4 sensors-22-05840-f004:**
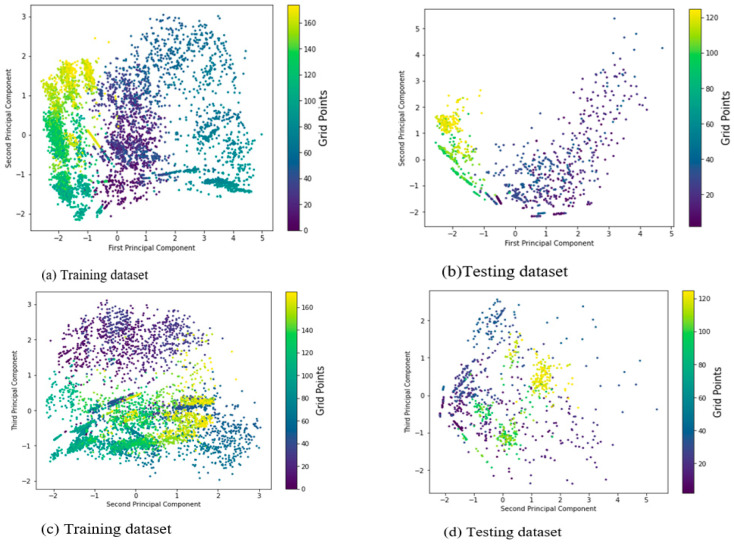
Distribution of principal components accounting for different EVRs with their corresponding labels for the dataset.

**Figure 5 sensors-22-05840-f005:**
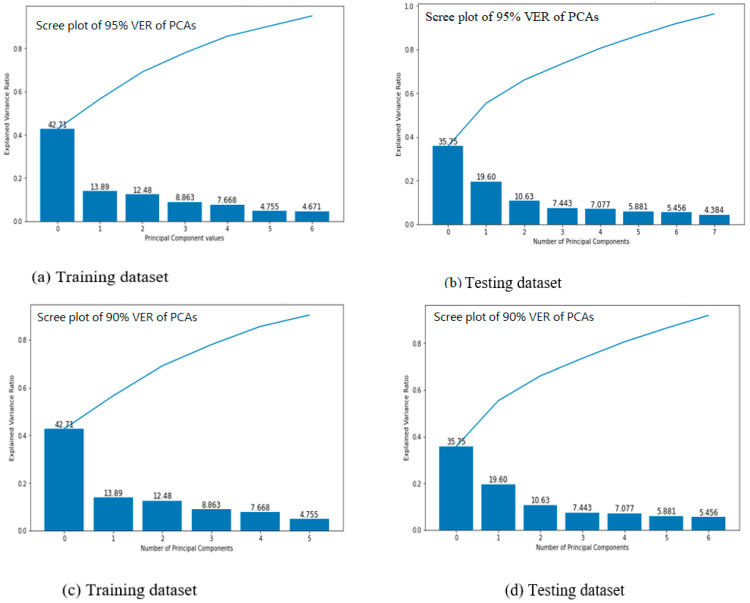
Scree plot of the 95% and 90% VER of principal components for the dataset.

**Figure 6 sensors-22-05840-f006:**
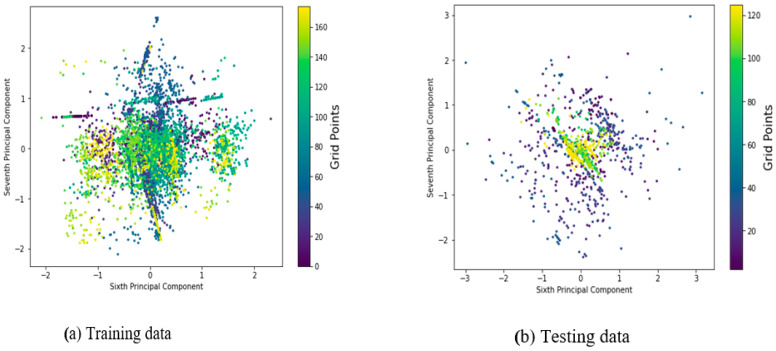
Distribution of principal components accounting for 20% EVRs with their corresponding labels for the dataset.

**Figure 7 sensors-22-05840-f007:**
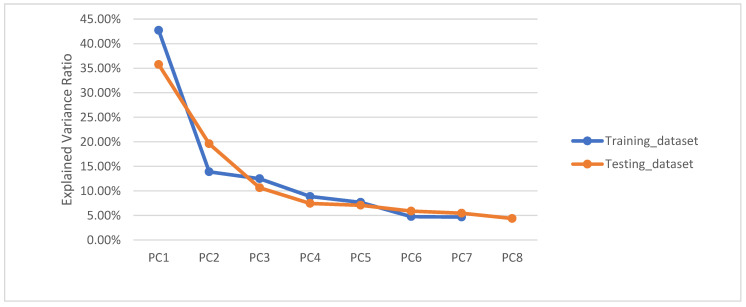
Contribution of each principal component to account for model variation for indoor positioning.

**Figure 8 sensors-22-05840-f008:**
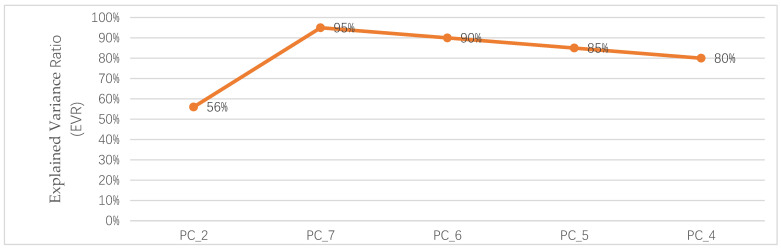
Contribution of the number of principal components to account for model variation for indoor positioning.

**Figure 9 sensors-22-05840-f009:**
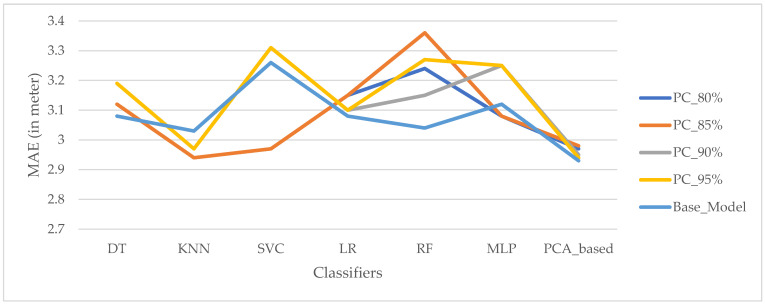
Effect of number of PCs for positioning with various explained variance ratios for the dataset.

**Figure 10 sensors-22-05840-f010:**
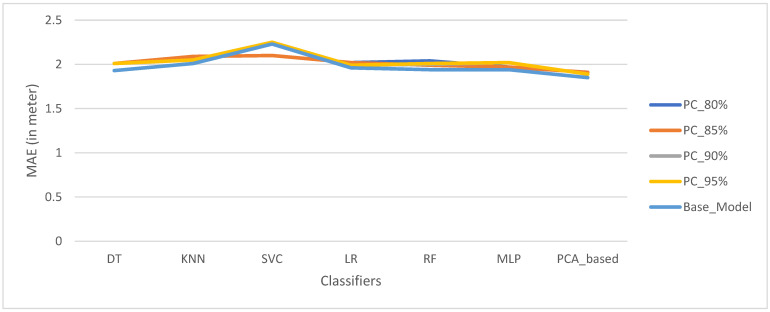
Effect of number of PCs for positioning with various explained variance ratios for the dataset after transfer learning.

**Table 1 sensors-22-05840-t001:** Effect of feature dimensions on the variance account of the predictive model for the dataset using training instances.

#PCA	Explained Variance Ratio (EVR)
2	56%
7	95%
6	90%
5	85%
4	80%

**Table 2 sensors-22-05840-t002:** Distributions of Wi-Fi received signal strengths per label for the dataset.

	Labels	134	10	127	-	--	51	59	101
# of RSS values per label	Dataset (#Labels = 175)	30	30	30			30	30	30

**Table 3 sensors-22-05840-t003:** Effect of feature dimensions on the variance account of the predictive model for both instances of training and testing dataset.

#PCA		Explained Variance Ratio (EVR)
Training Dataset	Testing Dataset	
2	2	56%
7	8	95%
6	7	90%
5	5	85%
4	4	80%

**Table 4 sensors-22-05840-t004:** Effect of each PCA on the explained variance account of the predictive model for the dataset.

PCAs Account for 95% Model’s Variation	Explained Variance Ratio (EVR) of Each Principal Component	
List of Principal Components	Training Dataset	Testing Dataset
PC1	4271%	35.75%
PC2	13.89%	19.60%
PC3	12.49%	10.64%
PC4	8.863%	7.443%
PC5	7.669%	7.077%
PC6	4.756%	5.881%
PC7	4.671%	5.456%
PC8	-	4.384%

**Table 5 sensors-22-05840-t005:** Correlation analysis of the features to see the effect of multicollinearity.

	X1	X2	X3	X4	X5	X6	X7	X8	X9
X1	1.000								
X2	0.4370	1.000							
X3	0.0701	−0.0439	1.000						
X4	−0.3447	−0.4317	0.0748	1.000					
X5	0.4292	0.2644	0.2307	−0.0714	1.000				
X6	0.4376	0.3687	0.0986	−0.0794	0.3869	1.000			
X7	−0.1143	−0.1789	0.1685	0.4785	0.0297	0.0457	1.000		
X8	0.5413	0.5607	0.1154	−0.5414	0.3588	0.3585	−0.2629	1.000	
X9	−0.2844	−0.3194	0.2860	0.4452	−0.0601	−0.0791	0.3008	−0.3028	1.000

**Table 6 sensors-22-05840-t006:** Model target’s positions using the various PCs for dataset PCA-based feature selection approach. Before TL: mean absolute error (MAE in meters). PCA’s explained variance ratio (EVR).

Classifiers	80% (#4)	85% (#5)	90% (#6)	95% (#7)	Base
Decision tree	3.12	3.12	3.19	3.19	3.08
K-neighbor (KNN)	2.94	2.94	2.97	2.97	3.03
Support vector machine (SVC)	2.97	2.97	3.31	3.31	3.26
Logistic regression (LR)	3.15	3.15	3.10	3.10	3.08
Random forest	3.24	3.36	3.15	3.27	3.04
Neural network (MLP)	3.08	3.08	3.25	3.25	3.12
The proposed algorithm-based PCA	2.97	2.98	2.95	2.94	2.93

% (#)---> represents PCA’s proportion of explained variational ratio (EVR) along with the corresponding number of feature spaces.

**Table 7 sensors-22-05840-t007:** Model target’s positions using the various PCs for dataset PCA-based feature selection approach. After TL: mean absolute error (MAE in meters). PCA’s explained variance ratio (EVR).

Classifiers	80% (#4)	85% (#5)	90% (#6)	95% (#7)	Base
Decision tree	2.01	2.01	2.01	2.01	1.93
K-neighbor (KNN)	2.09	2.09	2.05	2.05	2.01
Support vector machine (SVC)	2.10	2.10	2.25	2.25	2.23
Logistic regression (LR)	2.02	2.02	1.99	1.99	1.96
Random forest	2.04	1.99	2.00	2.01	1.94
Neural network (MLP)	1.97	1.97	2.02	2.02	1.94
The proposed algorithm-based PCA	1.91	1.91	1.89	1.89	1.85

***Base Model***---> represents a classifier having two principal components accounts for 57% of the total proportion of explained variational ratio (EVR), ***TL*** ---> transfer learning.

**Table 8 sensors-22-05840-t008:** Model target’s positions using the hybrid-based feature selection approach.

	Before TL	After TL
Classifiers	Mean Absolute Error (MAE in Meters)	
Decision tree	2.31	1.08
K-neighbor (KNN)	1.83	1.27
Support vector machine (SVC)	1.79	0.87
Logistic regression (LR)	2.53	2.06
Random forest	2.52	1.18
Neural network (MLP)	1.97	1.34
The proposed algorithm (hybrid based)	1.78	1.35

## Data Availability

The dataset used for this study are available upon request to the corresponding author.

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
