# Peer review of "Heterogeneous Transfer Learning for Wi-Fi Indoor Positioning Based Hybrid Feature Selection"

_sensors, 2022, doi:10.3390/s22155840_

Round 1

Reviewer 1 Report

This paper introduces two feature selection algorithms for WiFi fingerprinting-based indoor positioning.

The authors claimed that the proposed approach has greatly improved positioning estimation accuracy. However, there are some concerns about the paper which I will list below as I go through each section of the paper.

1)   Introduction

The authors may consider going straight to the main topic rather than explaining other indoor positioning techniques unrelated to the paper.

The authors should provide a sufficient explanation of the advantages of this research by explaining the disadvantages of all mentioned methods.

The authors should provide references for the statement ‘Several machine learning (ML) algorithms were also used to solve the RSS fingerprint-based indoor positioning problem, but … for positioning purposes.’

2)   Contributions

The authors may consider rewriting the contribution section 2 because belief is not a contribution. 

3)   Related work

The Related Work section should include more related content, e.g., feature selection for WiFi fingerprinting.

4)   Problem formulation and framework

The authors may consider moving the last two paragraphs above subsection 3.1 to Related Work.

Instead of using duplicated long sentences from the Introduction section, the authors may consider a clearer explanation of the purpose of PCA, Heterogeneous Transfer Learning, and objective function. 

The authors need to prove that PCA could avoid fingerprint duplication.

5)   Experimental Results and Discussion

The authors may reconsider the ‘sparse deployment’ of access points(AP), because there were 5APs deployed in the same line and all the APs were in the middle of the area.

The main purpose of subsection 4.2 is not clear.

The authors should explain the generalization of the complicated selection analysis of PCA features in subsection 4.2.

The authors should include a result comparison with other feature selection models to show the superiority of the proposed approach; because all included machine learning methods are only for prediction.

Subsection 4.5 should be moved to a different section because it is not an experiment result.

6)   Presentation

There are too many long sentences with grammar mistakes. 

There are numerous typos and grammar mistakes.

There are too many duplicated sentences used through-out the paper, even when explaining the main purposes of several proposed methods.

There is only one dataset but the authors used ‘1)Dataset A’.

Please use real nicely generated tables instead of screenshots.

Please reformat all the tables.

Figure 1 is poorly made and wrongly sized.

Formula 3 is oddly sized.

It is recommended that the authors address them in full, before the paper may be considered again for this journal publication. 

Author Response

We thank the Reviewer for the comments that helped us to improve the quality of the manuscript.

This paper introduces two feature selection algorithms for WiFi fingerprinting-based indoor positioning. The authors claimed that the proposed approach has greatly improved positioning estimation accuracy. However, there are some concerns about the paper which I will list below as I go through each section of the paper.

Comment #1:  Introduction

1a) The authors may consider going straight to the main topic rather than explaining other indoor positioning techniques unrelated to the paper. The authors should provide a sufficient explanation of the advantages of this research by explaining the disadvantages of all mentioned methods.

1b) The authors should provide references for the statement ‘Several machine learning (ML) algorithms were also used to solve the RSS fingerprint-based indoor positioning problem, but … for positioning purposes.’

Response #1a: Thank you for your valuable comment. We share the idea that we did not boldly stated as intend to be. However, in the current revised version of the manuscript, we have provided a sufficient explanation of the advantages of our research as per the suggestions we received. We have listed the most commonly used measurement methods for indoor positioning and carefully assessed what are the various possibilities in the literature before forwarding our proposed approach.

And 1b) we also provided references for the for the statement ‘Several machine learning (ML) algorithms were also used to solve the RSS fingerprint-based indoor positioning problem, but … for positioning purposes.’

Comment #2: Contributions

The authors may consider rewriting the contribution section 2 because belief is not a contribution. 

Response #2: Thank you for your comment. We have corrected it in the revised version and rewritten it as:

2) We applied correlation analysis (CA) to test independence of the Wi-Fi received signals from multiple access points mainly to reduce the effect of multicollinearity among the various RSS Wi-Fi access points aiming to avoid irrelevant features from the analysis. We claim that the RSS fingerprints collected from a grid point might be duplicated possibly from multiple Wi-Fi APs and create interference of matched patterns with the actual user’s fingerprint or irrelevant features might be generated and recorded in the database which negatively inflates or degrades the positioning performance.

Comment #3: Related work

The Related Work section should include more related content, e.g., feature selection for WiFi fingerprinting.

Response #3: Thank you for your valuable comment. We share the idea that we need to include more related content from various works and in the current revised version of the manuscript, we have included various works related to feature selection for Wi-Fi indoor positioning and categorically viewed as deterministic, probabilistic, and machine learning approaches.

Comment #4: Problem formulation and framework

4a. The authors may consider moving the last two paragraphs above subsection 3.1 to Related Work.

4b. Instead of using duplicated long sentences from the Introduction section, the authors may consider a clearer explanation of the purpose of PCA, Heterogeneous Transfer Learning, and objective function. 

4c. The authors need to prove that PCA could avoid fingerprint duplication.

Response #4a: Thank you for your valuable comments. We have moved the last two last two paragraphs above subsection 3.1 to Related Work. And we have consolidated as per the suggestion you gave us. Thank you for giving us an insight to improve our work.

4b. In the current revised version, we have provided a clear explanation of the purpose of PCA, Heterogeneous Transfer Learning, and objective function. 

4c. In the current revised version of our manuscript, we have explicitly described proving that PCA could avoid fingerprint duplication and we have added the following ideas in the revised version to describe the use of the PCA in our work and its details:

The proposed algorithm (feature selection based PCA or Algorithm 1)) is a pseudo code used to generate a new fingerprint feature space based on PCA. The input values are the original database fingerprints; after PCA is applied, a new fingerprint feature space with reduced dimension is created. Specifically, the feature selection based PCA algorithm applied data pre-processing techniques with the goal to reduce the dimension of the RSS measurements of the offline fingerprints database based on their contribution to the positioning performance. In other words, features with a higher explain-ability variance ratio are the more significant features that discriminate the model positioning performance and should be retained in the model for further analysis. The learned model is then applied during the testing phase after the PCA data preprocessing is performed to infer the mobile user's location based on the received new RSS measurements by mapping into the highest likelihood resemblance of signal-signature stored in the radio map. The feature selection based PCA algorithm primarily deals with the dimensionality curse of predefined radio map by linearly combining the features into an uncorrelated space (i.e. eigenspace) using the predefined radio map's training covariance matrix of size F x F. The fingerprint database or fingerprint matrix is projected into uncorrelated space in the direction of the highest variance ration and the Principal Components are chosen based on the highest explain-ability variance ratio (i.e. eigenvalue) which basically represents the information context of the principal components. Finally, algorithm 1 below also includes the detailed steps for PCA. Moreover, a real-life experimental dataset was used to validate our hypothetical claim. Thank you for giving us the insight to improve our work.

Comment #5: Experimental Results and Discussion

5a. The authors may reconsider the ‘sparse deployment’ of access points (AP), because there were 5APs deployed in the same line and all the APs were in the middle of the area.

5b. The main purpose of subsection 4.2 is not clear.

5c. The authors should explain the generalization of the complicated selection analysis of PCA features in subsection 4.2.

5d. The authors should include a result comparison with other feature selection models to show the superiority of the proposed approach; because all included machine learning methods are only for prediction.

5e. Subsection 4.5 should be moved to a different section because it is not an experiment result.

Response #5a: Thank you for your valuable comment. We share your idea still there is room to reconfigure the experimental layout in a different way to some extent. In our experiment, there were 9 Wi-Fi access points available to collect the received signal strength from a mobile device, allowing the fingerprint database to be created, and they were sparsely deployed to ensure that at least three access points (APs) could be detected at each grind point. And there may exit a chance to collect the RSS measurements from a grid point may be from more than three access points.

Response #5b: Thank you for your valuable comment. The main purpose of subsection 4.2 is to do an explanatory data analysis to better comprehend the distribution of the various features for establishing a possible meaningful insight and draw hypothetical claims which is a very important step before fitting the proposed model.

Response #5c: Thank you for your valuable comment. In the revised version, we have presented a detailed description of the PCA of features aiming to handle a dimensionality curse thereby enabling us to extraction of the most significant features for the modeling process.

Response #5d: Thank you for your valuable comment. We share your idea and this work presents two novel approaches for feature selection based on heterogeneous transfer learning applied to Wi-Fi indoor positioning. And we have identified various machine learning algorithms for their significant improvement in indoor positioning as cited in various research works. Hence, for consistency and validity of our experiments, we have applied those popular machine learning algorithms for their achievements both in prediction and in the felid of indoor positioning as a baseline for our work. Moreover, there are various works on indoor positioning using PCA techniques despite we did not get much based-on correlation analysis. However, the comparison they made was against machine learning algorithms. Furthermore, individual experimental results depend on the experimental setting and the environment. Thank you for giving us insight into improving our work.

Response #5e: Thank you for your valuable comment. In the revised version, Subsection 4.5 is moved accordingly to a separate section 5 as it is not an experimental result. Thank you for giving us insight into improving our work.

 Comment #6:  Presentation

There are too many long sentences with grammar mistakes.  Thank you for your valuable comment. In the revised version, we have corrected it.

There are numerous typos and grammar mistakes. Thank you for your valuable comment. In the revised version, we have corrected it.

There are too many duplicated sentences used throughout the paper, even when explaining the main purposes of several proposed methods. Thank you for your valuable comment. In the revised version, we have corrected it.

 There is only one dataset but the authors used ‘1) Dataset A’. Thank you for your valuable comment. In the revised version, we have corrected it to a dataset.

 Please use real nicely generated tables instead of screenshots. Thank you for your valuable comment. In the revised version, we have corrected it and all Tables are nicely generated as per your suggestion.

 Please reformat all the tables. Thank you for your valuable comment. In the revised version, we have corrected it and all Tables are reformatted and nicely generated as per your suggestion.

 Figure 1 is poorly made and wrongly sized. Thank you for your valuable comment. In the revised version, we have corrected it as:

Figure 1: Proposed Framework of Wi-Fi received signal strength-based Fingerprint for Indoor Positioning System

Formula 3 is oddly sized. Thank you for your valuable comment. In the revised version, we have corrected it as:

                    (3)

Comment #7:  It is recommended that the authors address them in full, before the paper may be considered again for this journal publication. 

Response #7: Thank you for your valuable comment. In the revised version, all comments and recommendations are well taken and addressed accordingly. We thank you for giving us insight for improving our work.

Reviewer 2 Report

This manuscript intends to present new methods to deal with the labour-extensive issues in using Wi-Fi for indoor positioning. This is a very interesting topic. If such issues can be solved, the indoor positioning based on Wi-Fi can go to another big stage. However, it seems that the authors did some contributions, but such things are not well presented. Therefore, I suggest a major revision. Some more detailed comments are the following:

1. The motivations and contributions of this research are not well presented. Please highlight them. For me, the manuscript does not clearly show the motivations. It seems that the authors are only happy with what they proposed only.

2. I think this manuscript is more of a report than a research paper. Because of the poor figures, tables, structures, I would say it is a report.

3. The resolutions of the figures are very bad and the figures have deformation. For example, in Figure 2, I cannot figure out what are the labels around room 2014.

4. There are many "on the other hand" without "on one hand". There are many "on the other hand", but you cannot even find a "on one hand".

5. The whole manuscript needs English editing. For instance, Line 312-315. "...to evaluate our proposed algorithms" This should be "...to evaluate our algorithms" or "...to evaluate the proposed algorithms". Another thing is "... dataset were first presented", in which 'first' should be 'firstly'. Furthermore, the manuscript is full of passive tones.

6. The figures and tables are screenshots from Word files. This is obvious and I do not think any explanations are needed. For instance, Figure 3, 4, 5, 6. Table 1, 2, 3, 4, 5, 6, 7, 8.

7. The structure of this manuscript is not good. For the time being, the methodology is mixed with experiments, which makes the experiments very heavy. I suggest extracting the methodology as a separate section.

8. Section 4.5 is not fitted to the Section of the experiment.

9. The line spacing in the manuscript needs to be adjusted. Whenever there is an equation, there are big spaces between line.

10. The formats of references need to be unified. By comparing the reference 1 and 2, we can find that the title of the reference 1 is surrounded by double quotes, that that of reference 2 is not, neither reference 31.

Author Response

Responses to the Comments of Reviewer 2

We thank the Reviewer for the comments that helped us to improve the quality of the manuscript.

This manuscript intends to present new methods to deal with the labour-extensive issues in using Wi-Fi for indoor positioning. This is a very interesting topic. If such issues can be solved, the indoor positioning based on Wi-Fi can go to another big stage. However, it seems that the authors did some contributions, but such things are not well presented. Therefore, I suggest a major revision. Some more detailed comments are the following:

Comment #1: The motivations and contributions of this research are not well presented. Please highlight them. For me, the manuscript does not clearly show the motivations. It seems that the authors are only happy with what they proposed only.

Response #1: Thank you for your valuable comment. We share your suggestion and we have made a major rewrite to have a better flow of ideas throughout the document. And we clearly presented the motivations in the revised version of the manuscript. We thank you for giving us an insight to improve the quality of our work.

Comment #2: I think this manuscript is more of a report than a research paper. Because of the poor figures, tables, structures, I would say it is a report.

Response #2: Thank you for your valuable comment. In the revised version, we have corrected it and all Tables and Figures are nicely generated as per your suggestion. And structures of the manuscript are properly adjusted. We thank you for giving us the insight to improve the quality of our work.

Comment #3:  The resolutions of the figures are very bad and the figures have deformation. For example, in Figure 2, I cannot figure out what are the labels around room 2014.

Response #3: Thank you for your valuable comment. In the revised version, we have corrected it and all Tables and Figures are nicely generated as per your suggestion.

Comment #4: There are many "on the other hand" without "on one hand". There are many "on the other hand", but you cannot even find a "on one hand".

Response #4: Thank you for your valuable comment. In the revised version, we have corrected it and removed it.

Comment #5:    The whole manuscript needs English editing. For instance, Line 312-315. "...to evaluate our proposed algorithms" This should be "...to evaluate our algorithms" or "...to evaluate the proposed algorithms". Another thing is "... dataset were first presented", in which 'first' should be 'firstly'. Furthermore, the manuscript is full of passive tones.

Response #5: Thank you for your valuable suggestion. We share your suggestion and we have corrected the writing style of the whole manuscript.

Comment #6: The figures and tables are screenshots from Word files. This is obvious and I do not think any explanations are needed. For instance, Figure 3, 4, 5, 6. Table 1, 2, 3, 4, 5, 6, 7, 8.

Response #6: Thank you for your valuable comment. In the revised version, we have corrected it and all Tables and Figures are nicely generated as per your suggestion.

Comment #7: The structure of this manuscript is not good. For the time being, the methodology is mixed with experiments, which makes the experiments very heavy. I suggest extracting the methodology as a separate section.

Response #7: Thank you for your comment. We shared your points as well and we have separated the methodology section from the experiment section. And section 4.5 has been moved to a different section named section 5. Thank you for giving us insight into improving the quality of our work.

Comment #8: Section 4.5 is not fitted to the Section of the experiment.

Response #8: Thank you for your valuable suggestion. We share your points and accordingly section 4.5 has been moved to a different section named section 5. Thank you for giving us an insight for improving the quality of our work.

Comment #9: The line spacing in the manuscript needs to be adjusted. Whenever there is an equation, there are big spaces between line.

Response #9: Thank you for your valuable suggestion. We have fixed it in the revised version.

Comment #10: The formats of references need to be unified. By comparing the reference 1 and 2, we can find that the title of the reference 1 is surrounded by double quotes, that that of reference 2 is not, neither reference 31.

Response #10: Thank you for your valuable comment. We share your points and we have corrected them accordingly. We thank you for giving us the insight to improve the quality of our work.

Reviewer 3 Report

1.    There are some repetitive parts in the Introduction, Related Works, and Problem Formulation and Framework sections. For example, lines 66-67-68 and lines 179-180-181. These and similar repetitive sections should be revised and omitted in the manuscript. The authors should carefully check the manuscript from top to down.  

2.    According to my view, the matrix of equation 3 looks like flatten and should be corrected.

3.    The image quality of Figure 1 is too poor and colorful. This figure should be rearranged. Figure 2 should also be refined. The term ELEVATOR is hardly readable on the figure.

4.    Due to absence of the any other Dataset, there is no need for naming the dataset as Dataset A. It is better to use only Dataset.

5.    In Figure 5, which graphics belong to 95% variance and which belong to 90% variance should be labeled to make them more understandable.

6.    The proposed hybrid approach that uses heterogeneous transfer learning and the implementation of transfer learning should be explained more in detail in the Experimental Results section.

7.    Figure 8 should be edited. There are five models in the legend and named in the figure. But in the figure, there is a nameless model, at the furthest right.

8.    All the figures and tables should be revised. Their dimensions should also be arranged. Some of them are horizontally stretched and deformed.

Author Response

Responses to the Comments of Reviewer 3

We thank the Reviewer for the comments that helped us to improve the quality of the manuscript.

Comment # 1: There are some repetitive parts in the Introduction, Related Works, and Problem Formulation and Framework sections. For example, lines 66-67-68 and lines 179-180-181. These and similar repetitive sections should be revised and omitted in the manuscript. The authors should carefully check the manuscript from top to down.

Response #1: Thank you for your valuable comment. We share your suggestion and we have made a major rewrite to have a better flow of ideas throughout the document. And we omitted the repetition of concepts within the manuscript. We thank you for giving us the insight to improve the quality of our work.
Comment # 2:  According to my view, the matrix of equation 3 looks like flatten and should be corrected.
Response #2: Thank you for your valuable comment. It is a good observation. In the revised version, we have corrected it as:

           --->         (3)
Comment # 3a: The image quality of Figure 1 is too poor and colorful. This figure should be rearranged. # 3b Figure 2 should also be refined. The term ELEVATOR is hardly readable on the figure.

Response #3a: Thank you for your valuable comment. In the revised version, we have corrected it as:

Figure 1: Proposed Framework of Wi-Fi received signal strength-based Fingerprint for Indoor Positioning System

Response #3b: Thank you for your valuable comment. In the revised version, we have tried but the term ELEVATOR is hardly readable still.

Fig. 2 Experimental layout conducted in UESTC for generating the dataset

Comment # 4:  Due to absence of the any other Dataset, there is no need for naming the dataset as Dataset A. It is better to use only Dataset.
Response #4: Thank you for your valuable comment. In the revised version, we have corrected it to a dataset.  

Comment # 5: In Figure 5, which graphics belong to 95% variance and which belong to 90% variance should be labeled to make them more understandable.
Response #5: Thank you for your valuable comment. In the revised version, we have corrected it as:

Fig. 5. Scree plot of the 95% and 90% VER of Principal Components for the dataset

Comment # 6:  The proposed hybrid approach that uses heterogeneous transfer learning and the implementation of transfer learning should be explained more in detail in the Experimental Results section.
Response #6: Thank you for your valuable comments. We share your suggestion and all issues are fixed in the revised version.

Comment # 7:   Figure 8 should be edited. There are five models in the legend and named in the figure. But in the figure, there is a nameless model, at the furthest right.

Response #7: Thank you for your valuable comment. In the revised version, we have corrected it as:

Fig. 8. Effect of Number of PCs for-positioning with Various Explained Variance Ratios for the dataset

However, Figure 9 depicts that the proposed algorithm-based PCA after applying transfer learning (HetTLIPPC based: heterogeneous transfer learning for indoor positioning based PCA) also reveals a significant positioning performance improvement and one can clearly observe that the proposed algorithm has come out to be the best algorithm with the lowest mean absolute error applied to the dataset with different principal components, accounting for 80%, 85%, 90%, 95%, and 56% variance explainability of positioning estimation. Surprisingly, the base model that constitutes approximately 56% of the variance explainability ratio of the model performs the best in both cases. However, due to the risk of model overfitting which is a very common phenomenon and is an imminent threat to indoor positioning as the dynamic nature of the indoor environment is described by NLOS propagation, multipath effects, and random signal noise. Thus, we recommend a model with a 95% variance of explain-ability ratio so as to represent the highest possible discriminant features which is more feasible in a complex indoor environment.

Fig. 9. Effect of Number of PCs for-positioning with Various Explained Variance Ratios for the dataset after transfer learning

Comment # 8: All the figures and tables should be revised. Their dimensions should also be arranged. Some of them are horizontally stretched and deformed.

Response #8: Thank you for your valuable comment. In the revised version, we have corrected it and all Tables and Figures are nicely generated as per your suggestion. We thank you for giving us the insight to improve the quality of our work.

Round 2

Reviewer 1 Report

The authors have addressed most of my concerns. The article may be now published.

Author Response

Notes on Revision

Re: Heterogeneous Transfer Learning for Wi-Fi Indoor Positioning based Hybrid Feature Selection

We would like to express our gratitude to the editors and the anonymous reviewers for their constructive suggestions and criticism. The comments are well taken, and the manuscript has been revised accordingly. Below please find our responses to the reviewers’ comments. Also, for the reviewers’ and editors’ convenience, major changes are written using MS Word track changes in the revised manuscript. And, we uploaded in the journal online system two files of a manuscript: In the second revised version, we prepared the files into Four formats. (i). A manuscript Word file with active track change as the Editors need it. (ii). A manuscript Word file of the same content after that accepts all track changes to avoid any technical issues related to the track changes. (iii). A manuscript Word file with a yellow mark is highlighted to show where the changes are made mainly to the reviewer as per his request. (iv). A blueprint of a manuscript Word file with PDF format-based Latex is also prepared to make sure that all mathematical equations and line spacing problems have been addressed.

Responses to the Comments of Reviewer 1

We thank the Reviewer for the comments that helped us to improve the quality of the manuscript.

In the second-round revision, we made significant changes in addressing the concerns raised in the previous version and we further enhanced the contents in all the issues of the sections raised by the reviewer including formatting issues, figures, and restructuring the manuscript as per the suggestions. Details can be found in the main manuscript file either the word or PDF final version can be seen.

Reviewer 2 Report

After comparing the reply file and the revised version, I still think this manuscript needs a major revision. For one thing, the authors submitted a perfunctory reply file and a revised version. Another thing is that the comments from reviewers are not well reflected in the revised version.

1. It is unacceptable that a revised version is in MS editing mode.

2. I cannot clearly see what has been improved. For example, the authors replied that "We share your suggestion and we have made a major rewrite to have a better flow of ideas throughout the document. And we clearly presented the motivations in the revised version of the manuscript. We thank you for giving us the insight to improve the quality of our work." I have no idea where are they. The highlighted texts?

3. Figure 2. Two reviewers said that the resolution is unacceptable. However, in the revised version, the authors only enlarged it. I still cannot figure out what are the labels around room 2104.

4. The authors said that they have corrected the writing style of the whole manuscript. However, such things are not clearly reflected in the revised version. I mean it seems that the authors only fixed some, rather than the whole manuscript.

5. The section 4.5 in the old version has been moved to a different section named section 5. I disagree with this revision because the contents of this section are methodologies rather than analysis.

6. I suggest the authors that please highlight where the revisions happen in the revised version. Meanwhile, please declare what has been revised and make an index there. For example, we revised the motivation. The changes can be found in the last paragraph of the introduction.

Author Response

Notes on Revision

Re: Heterogeneous Transfer Learning for Wi-Fi Indoor Positioning based Hybrid Feature Selection”

We would like to express our gratitude to the editors and the anonymous reviewers for their constructive suggestions and criticism. The comments are well taken, and the manuscript has been revised accordingly. Below please find our responses to the reviewers’ comments. Also, for the reviewers’ and editors’ convenience, major changes are written using MS Word track changes in the revised manuscript. And, we uploaded in the journal online system two files of a manuscript: In the second revised version, we prepared the files into Four formats.

(i). A manuscript Word file with active track change as the Editors need it.

(ii). A manuscript Word file of the same content after that accepts all track changes to avoid any technical issues related to the track changes.

(iii). A manuscript Word file with a yellow mark is highlighted to show where the changes are made mainly to the reviewer as per his request.

(iv). A blueprint of a manuscript Word file with PDF format-based Latex is also prepared to make sure that all mathematical equations and line spacing problems have been addressed.

Responses to the Comments of Reviewer 2

Comments and Suggestions for Authors:

After comparing the reply file and the revised version, I still think this manuscript needs a major revision. For one thing, the authors submitted a perfunctory reply file and a revised version. Another thing is that the comments from reviewers are not well reflected in the revised version.

  1. It is unacceptable that a revised version is in MS editing mode.

Response #1: Thank you for your valuable comment and we have prepared an option for the convenience purpose: please you can consider to use the iii. A manuscript Word file with yellow mark highlighted to show where the changes are made.

  1. I cannot clearly see what has been improved. For example, the authors replied that "We share your suggestion and we have made a major rewrite to have a better flow of ideas throughout the document. And we clearly presented the motivations in the revised version of the manuscript. We thank you for giving us the insight to improve the quality of our work." I have no idea where are they. The highlighted texts?

Response #2: We apologize for the inconvenience. In this revised current version, we have prepared an option that iii. A manuscript Word file with yellow mark highlighted to show where the changes are made. And all changes are heighted with yellow mark within the manuscript file.

  1. Figure 2. Two reviewers said that the resolution is unacceptable. However, in the revised version, the authors only enlarged it. I still cannot figure out what are the labels around room 2104.

Response #3: Thank you for your valuable comment. In the revised version, we have corrected it and all Tables and Figures are nicely generated as per your suggestion and Figure 2 is corrected as:

Fig. 2. Experimental layout conducted in UESTC for generating the dataset.

  1. The authors said that they have corrected the writing style of the whole manuscript. However, such things are not clearly reflected in the revised version. I mean it seems that the authors only fixed some, rather than the whole manuscript.

Response #4. Thank you for your valuable comment. We have corrected the writing style of the whole manuscript and significant changes are mainly made in the introduction, related works, results and discussions. We thank you for the constructive comments that greatly helped us to improve our work.

  1. The section 4.5 in the old version has been moved to a different section named section 5. I disagree with this revision because the contents of this section are methodologies rather than analysis.

Response #5. Thank you for your valuable comment. We have separated the methodology section from the experiment section. And section 4.5 has been moved to a methodology section named section 3.5 and found in Line 403-501. In the First-round revision, we placed it in a separate section 5. But, currently we placed it under the Methods section.

  1. I suggest the authors that please highlight where the revisions happen in the revised version. Meanwhile, please declare what has been revised and make an index there. For example, we revised the motivation. The changes can be found in the last paragraph of the introduction.

Response #6. Thank you for your valuable suggestion. As per your request, we have prepared a separate Word file, and please consider using this option available within the system: iii. A manuscript Word file with a yellow mark highlighted to show where the changes are made. We thank you for all the comments that greatly helped us.
